# Adult Neurogenesis of Teleost Fish Determines High Neuronal Plasticity and Regeneration

**DOI:** 10.3390/ijms25073658

**Published:** 2024-03-25

**Authors:** Evgeniya Vladislavovna Pushchina, Ilya Alexandovich Kapustyanov, Gleb Gennadievich Kluka

**Affiliations:** A.V. Zhirmunsky National Scientific Center of Marine Biology, Far East Branch, Russian Academy of Sciences, 690041 Vladivostok, Russia; ilyaak9772@gmail.com (I.A.K.); gleb.klyuka@bk.ru (G.G.K.)

**Keywords:** adult neurogenesis, traumatic brain injury, glutamine synthetase, neuroepithelial cells, neural stem progenitor cells, radial glia, cystathionine β-synthase, Pacific salmon, adult neural stem cells

## Abstract

Studying the properties of neural stem progenitor cells (NSPCs) in a fish model will provide new information about the organization of neurogenic niches containing embryonic and adult neural stem cells, reflecting their development, origin cell lines and proliferative dynamics. Currently, the molecular signatures of these populations in homeostasis and repair in the vertebrate forebrain are being intensively studied. Outside the telencephalon, the regenerative plasticity of NSPCs and their biological significance have not yet been practically studied. The impressive capacity of juvenile salmon to regenerate brain suggests that most NSPCs are likely multipotent, as they are capable of replacing virtually all cell lineages lost during injury, including neuroepithelial cells, radial glia, oligodendrocytes, and neurons. However, the unique regenerative profile of individual cell phenotypes in the diverse niches of brain stem cells remains unclear. Various types of neuronal precursors, as previously shown, are contained in sufficient numbers in different parts of the brain in juvenile Pacific salmon. This review article aims to provide an update on NSPCs in the brain of common models of zebrafish and other fish species, including Pacific salmon, and the involvement of these cells in homeostatic brain growth as well as reparative processes during the postraumatic period. Additionally, new data are presented on the participation of astrocytic glia in the functioning of neural circuits and animal behavior. Thus, from a molecular aspect, zebrafish radial glia cells are seen to be similar to mammalian astrocytes, and can therefore also be referred to as astroglia. However, a question exists as to if zebrafish astroglia cells interact functionally with neurons, in a similar way to their mammalian counterparts. Future studies of this fish will complement those on rodents and provide important information about the cellular and physiological processes underlying astroglial function that modulate neural activity and behavior in animals.

## 1. Introduction

Tissue regeneration in the vertebrate central nervous system (CNS) is a priority issue, encompassing new therapeutic strategies to help patients with neurodegenerative diseases or injuries [1]. The study of the new regenerative potential of neural stem progenitor cells (NSPCs) and their molecular control improves the prospects for therapeutic use while revealing the unique biology of the CNS in different vertebrate species [2]. In terms of regenerative capacity, there are significant differences between mammalian and non-mammalian vertebrate models [2,3]. In mammals, the limited tissue repair results from the absence of evident neuronal precursors, a generally unfavorable tissue environment for neurogenesis, the presence of glial/fibrillar scars, and the exacerbation of chronic diseases [3]. It is particularly interesting that mammalian astrocytes, oligodendrocyte progenitors, and ependymal cells can be converted into progenitors able to produce glia and neurons, as this suggests that they have the potential to repair neurons [4,5]. For example, it has been shown that inactive astrocytes can, under physiological conditions, be converted into neurons in vitro and in vivo by regulating levels of Notch signaling or inducing the expression of proneural factors such as Neurod1. This can serve as a source of neural repair after an acute traumatic brain injury [6,7,8]. In addition, ependymal cell populations in the forebrain and spinal cord of adult mammals also have the regenerative potential to generate neurons [9].

In contrast to mammalian models, the fish brain is considered to be a preferred vertebrate model for expanding our understanding of the cellular and molecular programs required for successful CNS regeneration [2,10]. In adult zebrafish, regeneration of the brain and spinal cord is driven by the presence of a variety of neural stem cells (NSCs) including, but not limited to, slowly and rapidly proliferating radial glia and neuroepithelial cells [11,12]. However, the full reparative potential of such cell populations, the set of neuronal lineages they can produce, and their intrinsic regulation in different stem cell niches remain unclear.

The study of different vertebrate models helps us to identify the unique characteristics of the central nervous system’s structure, thus allowing us to take a new look at adult neurogenesis [13]. It is important to study neurogenesis not only in model organisms, but also in salmonids who, unlike the model organism *Danio rerio* are, as a member of the advanced teleost family, phylogenetically old representatives who retain plesiomorphic features in the development of their brains [10]. The phenomenon of fetalization—demonstrating the delay of signs of embryonic structure in juvenile and adult animals—has been known for a long time and is characteristic of sturgeon (cartilaginous ganoids) and salmon fish.

## 2. Biological Features of NSCPs in Pacific Salmon

Studying the properties of NSPCs in a salmon fish model will provide new information about the organization of neurogenic niches containing embryonic and adult stem cells, reflecting their development, origin cell lines and proliferative dynamics. Currently, investigations of the molecular signatures of these populations in homeostasis and repair in the vertebrate forebrain are only at an early stage [3]. Outside the telencephalon, the regenerative plasticity of NSPCs and their biological significance have not yet been practically studied. The impressive capacity of juvenile salmon to regenerate brain suggests that most NSPCs are likely multipotent, as they are capable of replacing virtually all cell lineages lost during injury, including neuroepithelial cells (NEC), radial glia (RGC), oligodendrocytes, and neurons [14,15]. To date, this hypothesis appears to be supported largely by studies of the quiescent Müller glia in the adult human retina [16] and results from studies conducted on zebrafish [17,18]. However, the unique regenerative profile of individual cell phenotypes in the diverse niches of brain stem cells remains unclear. Various types of neuronal precursors, as previously shown, are contained in sufficient numbers in different parts of the brain in juvenile Pacific salmon [10,19].

Radial glia of the dorsal pallium in telencephalon has been the focus of most studies of CNS injury in zebrafish [2,20]. The properties of NSPCs should be studied dynamically, over different time periods, when the initial potential of NSPCs and their ability to participate in the reparative process in acute and/or chronic injury can be estimated [21]. The combination of experimental modeling of chronic and repeated acute injury provides an answer to the question of if high neuronal performance is maintained during repeated injury, which may determine the special properties of embryonic and adult NSPCs and neuronal precursors of Pacific salmon [22]. In the periventricular areas of the mesencephalic tectum, mesencephalic tegmentum, telencephalon and brainstem, high proliferative activity of cells has been established during acute injury (3–7 days post-injury) [14,23,24]. However, it is not yet known how the localization of neurogenic areas changes in the long term and how reparative programs are implemented when various parts of the salmon brain are damaged. Although the biological mechanisms associated with the high neuroreparative potential of juvenile Pacific salmon remain poorly understood, the significant sizes that Pacific salmon (such as chum salmon, sockeye salmon, coho salmon, and masu salmon) reach when feeding in the ocean [25] indicate that the increase in muscle mass is controlled by a vast number of neurons produced throughout life [26].

The multifold increase in the brain size of Pacific salmon is proportional to the enormous increase in their body length, which must be regulated by a highly differentiated central nervous system [10]. Chronically clarifying the mechanisms of the behavior of neuronal progenitors and/or NSCs after acute injury and changes in their activity may be a key to understanding the biology of different types of NSCs.

The largest number of primary neurodestructive changes, accompanied by a significant variation in calcium homeostasis, causing the activation of numerous intracellular enzymes, as well as the entry of extracellular glutamate and the development of excitotoxicity, occur after acute injury [14,24].

In salmon, glutamine synthetase (GS), an enzyme that neutralizes toxic extracellular glutamate, is synthesized in large quantities and after acute injury, and the number of GS+ cells in the area of injury and other areas of the brain increases [23,24]. It is possibly the enzyme that is associated with the successful process of post-traumatic recovery in the brain of juvenile Pacific salmon that ensures rapid and effective elimination of excitotoxicity and its replacement by other neuroprotective factors, particularly hydrogen sulfide (Figure 1).

An increase in the number of H_2_S-producing cells after injury in juvenile salmon also provides special conditions for neuronal repair, reducing the intensity of oxidative stress and normalizing the biochemical balance in brain tissue [27]. Glutamine synthetase and hydrogen sulfide, acting together through NMDA glutamate receptors [28], significantly contribute to reducing the toxic effects of glutamate and provide favorable conditions for the regeneration of nervous tissue, exhibiting natural neuroprotective properties. It is this feature of salmon that significantly distinguishes them from mammals, whose nervous tissue cells, after injury, produce an insufficient amount of these factors, which leads to the development of neuroinflammation [27]. It is also of interest that both markers are detected in cells of the proliferative periventricular zone (PVZ) of the salmon brain, which phenotypically correspond to NSPCs [10]. As a result of acute traumatic brain injury (TBI), the number of GS- and CBS-producing cells and the intensity of immunolabeling of these enzymes in these cells increases significantly, which indicates the direct involvement of these enzymes in the post-traumatic metabolism of NSPCs [14]. This observation differs significantly from the subgranular and periventricular zones of the mammalian hippocampus, where a decrease in GS production is observed in the cells of the corresponding zones, including aNSC [2,29]. The results of studies of traumatic brain injury in mice showed that the production of H_2_S increases within the first hours after TBI, but reaches the control level after 3 days and then decreases [28]. Taking into account that, under conditions of TBI, GS and CBS exhibit the properties of natural neuroprotectors that protect neurons from oxidative stress, we consider the increase in their production of salmon brain cells as additional factors that reduce the development of neurodestructive processes associated with injury.

Thus, it is obvious that NSPCs have a certain functional heterogeneity, and it is also clear that the synthesis of specific enzymes that help to reduce excitotoxic effects and oxidative stress in the NSPCs population of juvenile Pacific salmon indicates the activation of special genetic programs that are not characteristic of mammalian aNSCs.

The study of a large number of embryonic and adult-type precursors in the neurogenic regions of the brain located in the telencephalon, mesencephalic tegmentum, tectum, cerebellum and brain stem of juvenile Pacific salmon is suggested to be a representation of adaptive plasticity of the brain [23]. Juveniles in the first year are an extremely actively growing stage in the development of Pacific salmon. In natural conditions, they are preparing for the process of smoltification, accompanied by the activation of extremely adverse environmental conditions, including the transition from a freshwater environment to a marine one, which is in turn accompanied by the restructuring of all body systems [30,31]. In the life cycle of Pacific salmon, researchers distinguish four stages, which differ significantly from each other both in the combination and ratio of limiting factors, which affect each of them and the level of natural mortality [32]. Until recently, the most fully studied were the spawning, embryonic, and larval-fry (freshwater) stages, which collectively account for the greatest portion of losses in the formed abundance of generations [30]. Thus, for salmon with a short freshwater development cycle, such as pink salmon and chum salmon, these losses average 53.9% for the spawning stage, 39.6% for the embryonic stage, and 1.5% for the larval-fry stage [33]. Overall, from the embryonic stage to the larval-fry stage, the survival rate of these species is about 7%. Survival rates from the smolt to adult stages vary significantly among all species, ranging from 0.7% for chum salmon to 9.8% for coho salmon [34]. The highest and most variable level of mortality in the marine life-history period occurs in the early stages, during the coastal residence of juveniles, at the dawn of the study of the marine life-history period in salmon development [30].

The nervous system is a regulator of the functional activity of all organ systems of a growing juvenile salmon, and this stage of salmon development can therefore be considered a critical period of postembryonic neurogenesis, which is accompanied by the production of a huge number of embryonic-type neuronal precursors [10]. The latter are known to differ from RGCs, which include rapidly and slowly proliferating populations, and are the precursors of adult-type neurons; thus, in juvenile salmon in the first year of life, a huge reserve of future neurons is formed, which requires a thorough study [2]. Our current understanding of adult neurogenesis processes in various age groups of juvenile salmon, as well as in adult animals, needs further research to obtain a more balanced view and identify areas that require further research.

The ability to increase the production of new cells after brain injury can be controlled at the stem cell level or at the progenitor level [35]. Precursor ablation is the most common strategy to quickly replace lost cells and increase the size of the amplification pool. The use of progenitors is a hallmark of rapid growth and expansion of large brain regions such as the cortex or cerebellum during development but is also a key feature of adult ependymal/subventricular zone (SVZ) neurogenesis in rodents [36,37].

Results from zebrafish studies have shown that, during regeneration, NSPCs generally divide symmetrically, giving rise to two new neurons or two NSPCs [38]. Previously obtained data on juvenile Pacific salmon also indicate a significant predominance of NECs in the proliferative zones of the brain [14,19,24]. Thus, increased neurogenesis in response to TBI can rapidly deplete the stem cell pool unless the intensity of neurogenesis is carefully regulated. In this sense, long-term monitoring can show how the proliferative properties of NSPCs of juvenile salmon change with chronic TBI, as well as with a combination of chronic and repeated TBI [22].

Genetic and pharmacological data support the role of Notch signaling in the regulation of NSPCs quiescence in mammals and fish [6]. Specifically, in the zebrafish telencephalon, Notch3 is expressed by radial glia and promotes NSC quiescence [39]. This signaling occurs, in part, from progenitors [35]. Notch signaling appears to serve as an intrinsic signal to regulate quiescence and maintain stem cell populations [2,12]. Another challenging issue is to clarify the effect of inflammation on the processes of neuronal regeneration. Inflammation in the zebrafish brain is known to enhance the regenerative response [21]. However, data from mammals indicate that the occurrence of inflammation leads to the formation of stable neuropathological circuits, with their activity resulting in necrosis and subsequent replacement of the lost tissue with connective tissue scars [2].

## 3. NSCs in the Adult Brain

Stem cells are individual cells that have long-term self-renewal and can be multipotent. This conception generally corresponds to the classical scheme, according to which a stem cell generates another stem cell and differentiated offspring during division [40]. However, the tracking of clones in adult stem cell systems instead confirms a model in which stem cells are self-renewing and bipotent at the population level. In fish, adult neural stem and progenitor cells (aNSPCs) are mainly associated with the ventricular system [40]. In the fish telencephalon, aNSPCs have the typical morphology of radial glia and/or neuroepithelium and can be identified by several molecular markers of aNSPCs [41,42].

The persistent and widespread neurogenic activity of the adult brain in zebrafish was initially discovered using classical tracking techniques with thymidine analogues, leading to the identification of 16 proliferation domains that are present throughout the brain [43,44]. Similar methods, including those using BrdU (5-bromo-2′-deoxyuridine) and EdU (5-ethynyl-2′-deoxyuridine), which are incorporated into the DNA of cycling cells during mitosis, were applied; the neuronal precursors, physiologically silent but activated, were also identified in the spinal cord of adult zebrafish [45]. These constitutive and facultative neurogenic niches have generated great interest due to their diversity. They help to accomplish a comprehensive comparison of the identity and characteristics of neurogenic cells, as well as of the mechanisms underlying neurogenesis in the CNS of adult vertebrate species.

Other interesting techniques, such as retroviruses or DNA electroporation, have made it possible to identify the cells in contact with ventricles after intravevertebrate injection of a viral suspension [46]. After infectioning, viral genetic material is reverse transcribed and integrated into the host cell’s genome. Simple retroviruses can infect both dividing and non-dividing host cells whereas lentiviruses, for example, can penetrate nuclear pores [47]. The cellular specificity of gene expression is achieved through the use of specific promoters.

In DNA electroporation, the DNA, in the form of episomes, is introduced into the cells through injection or electroporation. This process reaches cells that are in contact with the lumen of the cerebral ventricles [48]. In this technique, the labeling of the cells is not permanent. The specificity of the labeling is regulated through the introduction of specific promoters. Some studies have shown that this method is particularly effective for cells with a large apical surface area [39].

In the pallium of adult zebrzfish, these properties are possessed by RGC [2]. In the pallial area of juvenile chum salmon *O. keta* and masu salmon *O. masou,* similar properties are characteristic of the NECs [14,49]. Additionally, it is possible to distinguish populations of constitutive and facultative NSCs, on the basis that the former are physiologically active, whereas the latter are usually silent, but activated, in case of injury, in the spinal cord of zebrafish [50] or the cerebellum of chum salmon [23]. Thus, the studies of various models suggest that aNSCs are a key element for the maintenance and repair of nervous tissue in adult organisms. These cells have the ability for self-renewal, and to differentiate and generate new neurons and glial cells, which is what makes them important for maintaining the nervous system function in normal circumstances. Further research that will enable better understanding of the mechanisms of NSCs and their role in the processes of tissue regeneration and repair after TBI is needed.

The existence of adult/newborn NSCs in the human brain [51] provides some hope for the use of these cells in regenerative therapy for neurodegenerative diseases or brain injury. Indeed, adult NSCs (aNSCs) produce new neurons involved in regeneration of damaged zebrafish brains, a regeneratively competent species [38]. Although aNSCs contribute to the renewal of mature neurons in some regeneratively competent species, early attempts to use endogenous aNSCs for repair of the mammalian brain unfortunately failed [52], apparently, due to a lack of understanding of the basic biology of aNSCs. Moreover, it also remains unclear to what extent the regeneration of the damaged brain requires changes in the behavior of aNSCs, compared to the intact brain, for the regeneration process to be successful. For example, regeneration of damaged cerebral cortex requires the generation of pyramidal neuron cells that are never generated by aNSCs in the undamaged brain [53]. It is therefore, important to compare cell behavior at the level of individual stem cells in the intact and damaged brains of both regeneratively incompetent and regeneratively competent species.

Studies of the brain of adult macaques have revealed an increase in the number of BrdU-positive cells in the SVZ after ischemia and a limited migration of these cells to the olfactory bulbs [54]. In another study, numerous BrdU-labelled cells were found in brain of adult squirrel monkeys (*Saimiri sciureus* brains) that were raised in enriched environments for three weeks without treatment [55]. BrdU positive cells were also observed in the ventral and ventrolateral striatum, including the nucleus accumbens. However, fewer of these cells were present in the caudal nuclei of the striatum.

Another study found that new neurons in the striatum of adult rabbits are created from progenitor cells located in the caudate nucleus. These cells express early neural markers, such as DCX (doublecortin), polysialylated neuronal cell adhesion molecule, class III β-tubulin, and HuC/D protein. Later, these neuroblasts migrate and differentiate into striated interneurons. Therefore, neurogenesis in the striatum of adult monkeys and rabbits does not depend on the neurogenesis of the adjacent SVZ (subventricular zone) [56].

The first steps towards understanding the cell behavior of individual aNSCs in neurogenic zones of the intact mammalian brain on the basis of lineage analysis [57] revealed the rapid consumption of a single aNSC, which produces heterogeneous neuronal output. However, various intracellular processes, such as cell death, selective proliferation, and terminal differentiation, may give way to the above-described features of adult mammalian neurogenesis.

The Cre-lox method, in being applied to genetic research, has provided a wealth of new and exciting information [58]. This technique, in being applied to transgenic animals that express Cre-ER under control of a transgene driver, uses precursor-specific neural promoters such as Her 4.1, Nestin, and GFAP. Nuclear translocation is temporally controlled by the administration of teamoxifen, which merges a reporter gene at lox sites, allowing strong expression of a reporter gene controlled by these promoters. This strong expression leads to effective recombination and labelling of progenitor cells and their descendants. Transgenic mice carrying the green fluorescent protein (GFP) gene were used to study striatal neurogenesis. They were injected with a recombinant adenovirus, which encodes Cre, into the lateral ventricle of the brain, which made it possible to specifically label SVZ cells and their offspring. The authors observed GFP-labeled cells that expressed DCX and NeuN in the striatum after the stroke [59].

In addition, the administration of epidermal growth factors (EGF) and fibroblast growth factor 2 (FGF2) into the lateral ventricles of adult rats has also been shown to increase their BrdU and NeuN levels following ischemia [60].

The use of Tet-rt TA—mediated genetic tracking also involves the use of double transgenic animals [20]. In this case, specificity of the induced cells is achieved using a specific Her4.1 promoter. Tet expression is controlled by neural promoters that are specific to precursors, and its activity can be temporally controlled by the administration of doxycycline. The label is temporary inside the precursor cells and disappears after cessation of doxycycline exposure. If the reporter protein is fused to histone (e.g., H2B), it will become diluted in progenitor cells during cell division, but remains stable in post-mitotic cells that are obtained shortly after induction. Therefore, this method can be used for the birth dating, similar to using thymidine, which causes loss of rapidly dividing cells as the label becomes diluted. Various degrees of induction can be used, ranging from weak (clonal) induction to complete induction. Complete induction can also be used to track non-dividing progenitor cells that do not retain a specific marker; this, however, should be done with caution, as the expression levels during induction may vary.

Adult neurogenesis has been confirmed in the human striatum using a technique that allows retrospective determination of the cell birth date in the body. The technique, developed by Ernst and his team, is based on the detection of changes in the level of the carbon-14 isotope in the DNA of proliferating cells by using the accelerator mass spectrometry. The levels of carbon-14 in genomic DNA closely correspond to those in atmospheric levels created during the Cold War period, making it possible to determine when DNA is synthesized in adult brain areas and the cell originates [51]. Ernest and his colleagues have determined that the cells of the human striatum are free from the age-related pigment lipofuscin, which suggests they represent young neurons. They also examined the patients with Huntington’s Disease (HD) and noted that postnatal neurons in the striatum of these patients were depleted at an advanced stages of the disease [51].

In addition, transcriptome data from adult brains has shown that DCX is mainly expressed in the striatum of adults, rather than in the hippocampus. This finding has been confirmed using other methods, such as Western blot analysis and immunohistochemistry by using different neuroblast markers, such as PSA–NCAM, in the human post-mortem brain. The authors found the same number of neuroblasts in the striatum, subventricular zone (SVZ), and hippocampus of human brain [61].

## 4. Creation of aNSCs in Intact and Damaged Brains

The regenerative capacity of the central nervous system largely depends on the number and development potential of NSCs. Since the developmental potential and neurogenic competence of NSCs in mammals declines during development in most CNS regions, the extent of functional and structural recovery of the CNS is also low [62]. In the adult mammalian CNS, despite the presence of a limited number of NSCs in all ventricles along the cerebral axis, most of them are latent. The exception is NSCs, which are localized in the subventicular zone (SVZ) of the lateral ventricles and the subgranular zone (SGZ) of the hippocampus. In these two zones, NSCs continuously generate new neurons that are functionally integrated into existing neural circuits.

However, recent research has shown that new neurons have been discovered in other parts of the adult mammalian brain, including the hypothalamus, striatum, substantia nigra, cortex and amygdale [63]. Some studies suggest that these new cells originate from pools of endogenous stem cells in these brain areas [64]. However, other data [63] suggest that the presence of these new neurons is the result of their migration from the subventricular zone (SVZ).

Proliferation of NSCs in the SVZ of the lateral ventricles leads to the emergence of GABA–ergic neurons, which then migrate to the olfactory bulbs throughout life [63]. In the SGZ, granular neurons are formed, which are then integrated into the *dentate gyrus* of the hippocampus. In the adult brain, both latent and active progenitor cells, including stem cells, can be mobilized to generate new neurons and glia in response to brain damage such as ischemia, TBI, and demyelination [64]. In Huntington’s disease, GABA–ergic neurons in the striatum are primarily affected. Other areas of the brain, particularly the hippocampus, become damaged in the later course of the disease [65]. Additionally, there is a general decrease in neurogenesis in the hippocampus [66], which occurs against the background of increased cell proliferation in the subventricular zone (SVZ), with their subsequent migration to the striatum [67]. This increase in neurogenesis in SVZ may serve as an endogenous compensatory mechanism that may provide an opportunity to slow down the disease progression.

Although it has been established that neurogenesis occurs in adults in various brain regions, the physiological significance of newly formed neurons in these regions, as well as the extent and significance of adult neurogenesis, has still not been fully understood [68].

Returning to research on fish, a significant number of nestin-positive precursors were detected in adult trout, in the *Oncorhynchus mykiss*, telencephalon [69], optic tectum [69], cerebellum and *medulla oblongata* of the brain. These findings indicate the neuroepithelial nature of NSCPs, as they show that NSCPs is present in these regions of the brain, similar to its presence in neural precursor cells.

Immunolabeling of doublecortin (DCX) in the dorsal matrix zone (DMZ) of the cerebellum (Figure 2A), in the surface zones of the Molecular layer (Figure 2A), projection cells of the Ganglion layer (Figure 2A) and projection cells of the Basal Zone (Figure 2B) in the cerebellum of adult trout clearly indicates different areas of localization of newly created cells, both in traditional presence of DCX+ cells in the neurogenic zone of the optic tectum (Figure 2C) and the *torus longitudinalis* (Figure 2D) of adult trout. This indicates the replenishment of the mesencephalic tectum with new neurons, which is the highest integrative center that regulates visual function and coordinates motor activity in conjunction with visual signals [69].

Unlike in mammals [70], neurogenic areas in adult zebrafish are distributed throughout the brain and are accessible for imaging. This is particularly pronounced in the dorsal telencephalon [44]. Intravital imaging provides information about changes in the behavior of aNSCs and their descendants in the regenerating and intact brain [38], which helps to study processes of aging and regeneration. In adult zebrafish, the pallium contains RG-like aNSCs that are capable of generating new neurons. Radial processes of aNSCs span the brain parenchyma, contacting with the basal membrane [38]. The morphology and antigenic profile of aNSCs in the zebrafish pallium resemble RG cells in the developing mouse telencephalon [71]. New neurons produced by pallial aNSCs in the intact zebrafish brain do not migrate but are stored in the progenitor cell zone [51]. These new neurons are interspersed with rapidly dividing progenitors that do not have the characteristics of stem intermediate progenitor (IP) cells or nonglial progenitors [47]. During TBI, a special aNSC program is activated, which leads to the formation of new neurons that migrate to damaged areas of the brain [72]. Of particular importance is that injuries such as small puncture wounds induce reparative neurogenesis without affecting ongoing neurogenesis in the uninjured brain. Thus, issues about the origin of these new neurons and their relationship with the normal generation of adult neurons are not fully clarified. In adult trout, DCX+ cells were found in the subpallium (Figure 2E), which corresponds to data obtained for zebrafish [2]. However, it is still unknown if these neurons are formed in adulthood and survive.

In contrast to the pallium, the neurogenic niche in the zebrafish subpallium is more similar to the mouse subependimal zone (SEZ), as the number of IPs is proportionally greater [73], with some of them also migrating to the olfactory bulbs. Of certain interest is the fact that in the zebrafish subpallium, aNSCs have low levels or do not express glial markers because they express nestin and the factor ZO-1 [73], thus resembling neuroepithelial cells. The mixture of radial and tangential migration of NSC progeny in this region makes clonal analysis difficult. Thus, the behavior of subpallial aNSCs in zebrafish at the single cell level remains to be assessed.

Studies of the aNSCs behavior have sought mechanisms to increase the output of neurons for regeneration without affecting constitutive neurogenesis [38]. The increase in neuronal production can be explained by several mechanisms, such as increased proliferation of aNSCs and nonglial progenitors and decreased cell death. In this case, the expression of some transcription factors increases, and the Pax family in particular [74]. However, in adult trout, the level of Pax6 expression in neurons and radial glia of the hypothalamus is quite high (Figure 2F). Studies have shown that the number of both aNSCPs and non-glial precursors increases after the brain injury [75]. However, it has not yet been possible to observe several successive divisions of single aNSCs [75]. Some aNSCs that divide after injury do not self-renew, but rather are exhausted due to symmetric divisions giving rise to two non-glial progenitors, which results in the generation of more neurons from a single aNSC. Similar exhaustion of aNSCs is observed in a mouse *dental gyrus* (DG) model of neuronal hyperactivity, which leads to a decrease in the ability of the zebrafish brain to recover from damage caused by repeated trauma [76]. Alternatively, zebrafish contains more primitive, undifferentiated aNSCs [77] with the ability to replace depleted aNSCs after injury. Thus, issues include the possibility of repopulation of the neurogenic zone and tissue restoration after several injuries and differences in the regenerative ability of organs depending on their type.

In adult neurogenesis, some NSPCs are limited to SVZ and migrate to the damaged striatum in mammals and humans before being differentiated into interneurons [13]. However, few studies have explored the molecular mechanisms underlying neurogenesis induced by pathological conditions that potentially regulate proliferation, migration, and differentiation in the striatum.

Studies by Lee and colleagues in 2021 described similar epigenetic mechanisms involved in neurogenesis after Alzheimer’s disease (AD) in adults by investigating the effects of suppression of AD7c–NTP after AD injury. The AD7c–NTP is associated with neurodegeneration in AD, and suppression is partially mediated by the phosphorylation of MeCP2 at serine 421 (S421), in combination with DNA demethylation at GFAP, Nestin, and DCX promoter regions, which prevents MeCP2 from binding to its target cells and thus reducing transcription repression, and in turn induces gene expression. These genes may be involved in regeneration and determination of the fate of NSPCs (neural stem/progenitor cells) during striatum neurogenesis in adults [78].

Continuous improvement of research methods in this area is crucial. One possible way to further clarify the degree of neurogenesis in the adult brain is to use positron emission tomography (PET). However, the prospect of generating new neurons in the human brain, which is relevant to chronic neurodegenerative disorders, acute neurological conditions, and metabolic diseases, as well as treatment of these conditions, is an interesting area for research [79].

Neurogenesis in the adult striatum is typically limited under normal physiological conditions, but can be triggered by a variety of procedures, such as in response to pathological stimuli, such as a stroke/ischemic event or trauma, or pharmacological stimuli, such as administration of certain growth factors or neurotrophins [80]. This process has been studied in animal models and humans. Several studies have identified two potential sources of newly formed neurons in the striatum: the precursors of the subventricular zone (SVZ) and local precursors within the striated parenchyma. It has been proposed that these neurons migrate from the SVZ to the striatum, while the latter are progenitor cells activated by neurotrophic factors to become neurogenic. This adult neurogenesis may be a mechanism by which the brain attempts to recover from injury [81].

Thus, the function of striatum neurogenesis in adults has yet to be established. The longevity of neurons born in adults indicates a possible functional integration that can be used for therapeutic purposes in patients with striatum disorders such as Huntington’s disease, Parkinson’s disease (PD), Alzheimer’s disease (AD) and other disorders [80].

## 5. A Comparison of Functional, Structural and Physiological Properties of RGCs in the Zebrafish Brain and Mammalian Glia

Zebrafish has a high neurogenic and regenerative potential and is thus a well-established experimental model of choice for studying neural stem cells and neurodegenerative diseases [1,2,11,17,20,21]. Therefore, we review the cellular and molecular aspects of the neurogenic potential of zebrafish radial glia, and then compare them with those of mammalian radial and astroglia.

During vertebrate neuronal development, the neuroepithelium differentiates into glia as the ventricles develop [82]. The earliest form of glia is radial glia which, as the name suggests, extends from the apical end of the ventricle to the basal layer and is often adjacent to blood vessels. Radial glia act both as NSCs that give rise to neurons and intermediate progenitors and as scaffold cells along which new neurons migrate [83] and form cortices [84,85]. Thus, dysfunction of RGCs leads to neurodevelopmental disorders, indicating the primary importance of glia for brain development.

During the late embryonic and early postnatal periods of development in mammals, radial glia detaches from the ventricular region and retracts its processes to form a stellate structure in the parenchyma. These cells are referred to as astrocytes or astroglia, and the process is referred to as astrogliogenesis [86]. The neurogenic capacity of glial cells decreases as they transition from radial glia to astrocytes [87].

During postnatal stages in mammals, most astroglia cells are nonneurogenic, although astrocytes divide, expand, and populate almost the entire central nervous system [88]. However, some astrocytes are deposited in specific brain regions and continue to be neurogenic progenitors throughout life [89]. Two such regions have been well studied, which are located in the telencephalon, and referred to as neural stem cell niches of the subventricular zone (SVZ) of the lateral ventricle and the *dentate gyrus* of the hippocampus [90,91]. In humans, adult neurogenesis is still controversial, but many studies support the presence of newborn neurons in the adult brain [92,93]. These neural stem cell niches generate neurons with limited lineages but are insufficient to generate sufficient numbers of cells to restore the original functionality of neuronal networks in some diseases [94,95]. In fact, neural stem cells and neural progenitors even decrease their proliferative and neurogenic capacity in many brain diseases [95], which may explain why the mammalian brain regenerates poorly, and activation of the neuroregenerative abilities of neural stem cells is emerging as a potential way to combat various neurological disorders [96,97].

However, other molecular mechanisms are not well understood but are important because they regulate the transient increase in NPCs proliferation in the SVZ and the migration of neuroblasts to damaged areas after stroke [98]. These endogenous negative regulators include the LNK protein, which is expressed in NPCs in the SVZ of adult rodents and humans. When the LNK protein is expressed at lower levels, NPC proliferation increases in the SVZ due to the activation of STAT1/3 transcription factors. When the LNK protein is overexpressed, however, it weakens the signaling pathway for insulin-like growth factor 1 (IGF1) by inhibiting the phosphorylation of AKT, which leads to a decrease in NPC proliferation. Another endogenous negative regulator of NPC proliferation induced by stroke in the SVZ is the tumor necrosis factor-receptor 1 (TNFR1). An increase in cell proliferation within one week of stroke is associated with increased numbers of microglial cells and the expression of genes for TNFR1 and tumor necrosis factor (TNF)-α in the SVZ. Blockade of TNF–R1 signaling can promote cell proliferation in the SVZ and enhance the formation of neuroblasts [98].

Inflammation is a rapid response to injury, involving the activation of different types of immune cells [13], which can be sensitive, local, and tissue-resistant. They can also self-renew, as in the case of microglia in the central nervous system. Additionally, macrophages can originate from peripheral monocytes. Neutrophils and other cells of the immune system, such as B and T cells, also play a role in inflammation. Circulating immune cells, such as monocytes, macrophages, and granulocytes, can attract additional immune cells to the injury site through the secretion of inflammatory mediators, in a process known as chemotaxis [13].

In mice, after injury to the cerebral cortex, there has been accelerated recovery of functional behavior [99]. These findings suggest that secondary injuries, such as edema, metabolic disorders, and reactive oxygen species produced during traumatic damage to the central nervous system, may also play a role in poor recovery. However, a new type of dormant neural stem cells has been recently identified, which are activated but still maintain their resting state after TBI. These cells can be specifically activated by interferon–gamma (IFNg) signaling after ischemic damage [100]. Interestingly, in non-mammalian vertebrates, where regeneration occurs effectively, there is a rapid inflammatory response. This regenerative potential, despite the strong inflammatory response, appears to differ from that of mammals, where severe, and often persistent, inflammation and the formation of scar tissue from reactive astrocytes are considered some of the major obstacles to successful regeneration [13,101].

In zebrafish, the progression of neuroepithelium to astroglia and oligodendrocytes predominates as the fish grows. However, the degree of astrogliogenesis is lower, and glia cells retain their identity as radial glia throughout life [12,72]. This provides a larger pool of RGCs and neuroepithelium during adult stages and may explain why neurogenic capabilities are more widespread in zebrafish, compared to in mammals. Zebrafish contains NSC niches that are distributed throughout the rostrocaudal axis, which is limited by the ventricles [102,103]. In many of these brain structures, newborn neurons are sequentially organized into layers, as described in the telencephalon [20], habenula, and spinal cord [104].

Several regions of the adult zebrafish brain have been studied to ascertain their neurogenic abilities [12,72]. The most widely studied NSCs are in the dorsal telencephalon of adult zebrafish [105,106,107], which consist of radial glia that, when dividing, express proliferating cell nuclear antigen (PCNA) and Mcm5 [106]. This proliferation of RGCs is referred to as constitutive proliferation because Alzheimer’s disease continues throughout the life of organisms, although its symptoms decrease with age and lead to a homeostatic regime of neurogenesis [12,107]. Notch signaling and associated molecular cascades, are among the major determinants of neurogenesis, regulating the balance between quiescence and proliferation [104,105]. In addition to Notch, hyperglycemic states and hormonal signaling through estrogen have also been shown to maintain the stability of the radial glial stem cell pool [108,109]. Overall, it appears that quiescence in the adult zebrafish brain is an important way to maintain the proliferative capacity of neural stem cells throughout life, as has also been shown for the mouse brain [110,111,112].

## 6. Constitutive and Reparative Neurogenesis in the Pacific Salmon Brain

During their life cycle, Pacific salmon undergo a number of significant physiological changes. Juveniles during smoltification adapt to varying salinity when moving from fresh water into the ocean [32]. In the first year of life, the juveniles grow especially actively, showing a tremendous increase in body weight [25]. In this regard, the fact of a multifold increase in the brain volume of juvenile salmon is interesting and intriguing. Indeed, in the first year of life, the weight of the salmon brain increases many times over, as a result of a huge increase in the number of new cells that appear as a result of the proliferative activity of the matrix zones of the brain containing neural NSPCs.

The homeostatic activity of the NSPCs in certain periventricular matrix zones of the brain located in the integrative centers (such as telencephalon, cerebellum, optic tectum, and brain stem) causes a huge number of new neurons to appear in the first year of life of juvenile Pacific salmon (Figure 3). Some of these cells with high proliferative potential are apparently capable of migrating into the brain parenchyma from the primary proliferation zones and can form centers of secondary neurogenesis (Figure 3). Such properties are largely specific to salmon and are not found in other fish species. Apparently, salmon aNSPCs are highly plastic and regulated by heterogeneous genetic programs, through which they are activated as a result of homeostatic growth or a post-traumatic impulse, possessing the properties of embryonic-type NSCs that divide through symmetrical mitoses (Figure 3). This is especially clearly seen in acute damage to the medial tegmentum of juvenile chum salmon *Oncorhynchus keta* [19]. In this area, other fish, particularly Danio, exhibit practically no neurogenesis. However, after TBI in juvenile chum salmon, the accumulation of HuCD+ neurons was detected directly at the injury site and at the base of the injury [24]. They are capable of migration and constitute the major portion of new neurons in the brain of juvenile salmon. Newly formed cells are also distinguished by functional heterogeneity. A part of the population retains embryonic properties, is capable of active proliferation, and contains the embryonic NSC markers vimentin, nestin, and aromatase B [14,19].

Another population of NSCs is detected in older individuals. For example, in trout, they are radial glial cells expressing glutamine synthetase [69]. The presence of a large number of GS+ RG cells in the telencephalon, optic tectum [69], and the *medulla oblongata* (unpublished data) convincingly indicates the predominance of adult-type aNSPCs in salmon aged 3 years and older, compared with younger stages of Pacific salmon postembryonic development in younger stages of Pacific salmon postembryonic, when NE type cells predominate in similar brain centers [14,19,24]. It is of particular interest that RG cells begin to express GS, GFAP and CBS in some areas of the pallium and subpallium of juvenile masu salmon *Oncorhynchus masou* and chum salmon *Oncorhynchus keta* in response to TBI [14,24,49]. These data contrast with the results of studies of intact juvenile *O. masou* and *O. keta*, where GS, GFAP, and CBS were expressed by NE type cells in most pallial and subpallial zones of the telencephalon [14,24,49].

The mechanisms of neuroprotection in various vertebrates, including mammals, are conservative. For example, in TBI, glutamine synthase is involved in converting toxic glutamate into neutral glutamine and reducing exitotoxicity, as well as regulating the processes of neurogenesis, including the formation of new neurons and their introduction into existing neural networks in the mammalian hypocampus [113,114]. Another enzyme, aromatase B, catalyzes the conversion of testosterone to estrogen, playing a key role in the synthesis of estrogens that are involved in the brain recovery after injury [115]. Hydrogen sulfide detected in the brains of Pacific salmon can act on a number of molecular targets, such as cytochrome C and caspases, to prevent programmed cell death. In addition, H2S promotes neurogenesis, which is also an important aspect of brain repair [116,117]. Thus, inhibitors or activators of neuroprotective factors detected in the brains of Pacific salmon may become targets for the development of new therapeutic strategies.

RG cells are distinguished by heterogeneous proliferative properties (only a part of the population is capable of proliferation, and many cells remain in a “silent” state) and heterogeneous molecular phenotypes that allow effective restoration of damaged tissue after traumatic injury (Figure 3). As a result of a long post-traumatic period in the juvenile salmon brain, the area of injury almost completely regenerates due to cells migrating to the damage zone and also to the primary proliferation zone (PVZ) and submarginal zone (SMZ) that contains both neuroepithelial-type eNSCs (Figure 4) and RGCs (Figure 4).

Thus, juvenile Pacific salmon are a unique model with enormous neurogenic potential that has not yet been well studied to date. Since the nervous system is the main regulatory system of the body, the potential for the Pacific salmon CNS to retain the features of embryonic structure (a phenomenon known as *embryonalization*) is of great theoretical interest, and is also essential for studies of the biology of vertebrate NSCs. The need to actively study the properties of aNSPCs is determined by the existence of heterogeneous activation programs of reparative scenarios implemented in the brain of juvenile Pacific salmon that allow these animals to survive in the most adverse ocean conditions. To date, data on how aNSPCs of juvenile salmon behave under conditions of long post-traumatic period remain scarce [19,49]. Little is also known about how the properties of aNSPCs change with repeated damage to the CNS structure [27]. The question of if the ability for successful repair in aNSPCs is retained after repeated TBI in juvenile salmon is intriguing and requires detailed study. A combination of experimental modeling of chronic and repeated acute injuries can give a clue as to if the high performance of neurons during a repeated traumatic process is maintained or, *vice versa*, decreases, which can in turn verify the characteristics of embryonic and aNSCs in Pacific salmon.

By expanding our comparative analysis using diverse vertebrate models, particularly those with limited regenerative capabilities, we can further emphasize the significance of research findings related to Pacific salmon and clarify potential evolutionary paths or mechanisms that lead to these differences.

Pacific salmon belong to a group of ray-finned fishes, a branch of which emerged on the evolutionary tree rather recently, approximately 400 million years ago. One of the most intriguing features of the salmon family is the full-genome duplication event, designated as SsR4 or WGD4, which occurred approximately 70 to 80 million years ago, according to [118]. This event has occurred at various times, ranging from 96 ± 5.5 million years ago to the present day, as reported by [119].

Genome-wide duplication plays a significant role in the evolution of vertebrates, particularly in fish species. It contributes to the development of new functional capabilities by copying not only structures but also regulatory regions of genes. At the same time, half of the resulting tetraploid genome continues to control previous functions that support current adaptations, while the other part is freed from those functions and can rebuild quickly. This process is not limited by stabilizing selection, which determines gene complex differentiation and evolutionary adaptation to the environment, and directly affects speciation processes, as described by Ohno [120] and further explored by Se’mon and Wolfe [121].

The evolutionarily plastic base created by the full-genome duplication of WGD4, under conditions of significant heterogeneity and temporal variability of the habitat, ensured the emergence of large taxa of salmonids and their further divergence [122]. Genetic doubling of the genome has led to the emergence of autotetraploids [123], which lack paired conjugation and segregation of chromasomes; full-genome duplications are therefore accompanied by so-called gene fractionation (i.e., a decrease in the proportion of duplicated functional sections of the genome that turned out to be superfluous in a certain sense) and re-diploidization (i.e., the formation of normal chromosomal bivalents in meiosis). In addition, the presence of isolocuses is noted in Pacific salmon, both in polymorphic proteins and enzymes [124] and in amplified DNA fragments [125], which indicates the presence of identical pairs of loci. In evolution, an undoubted role was played not only by full-genomic, but also by other mechanisms of duplication and other genomic rearrangements that ensure a change not of the entire genome, but of its part [126], including unequal crossing (between sister chromatids or homologous chromosomes) or the occurrence of multiple tandem copies in the process of the DNA copying, which has constantly occurred in the evolutionary history of species.

Thus, WGD4 was the last, but not the only, full-genome duplication in the phyletic lineage leading to salmonids in the evolutionary past. These data allow us to conclude that genome duplication in salmon fish may be one of the reasons for the high neurogenic activity of their central nervous system and the presence of centers of additional proliferative activity, not only in the territory of matrix periventricular zones, but also in the surface localizations and parenchymal neurogenic niches absent in other fish. Given the above, further identification of signaling pathways, transcription factors, and environmental signals that control these processes in Pacific salmon may provide a deeper understanding of the fundamentals of their ability to regenerate. In particular, studies of localization of Pax family transcription factors in the brain of adult, intact trout *Oncorhynchus mykiss* have found, after acute mechanical injury to the eyes, Pax2+ transcription factor distributed in the optic nerve of the trout [127].

Damage to the optic nerve in the brain regions of trout with directed retinal input—the diencephalon and the visual tectum—leads to an increased number of Pax2+ reactive astrocytes. These cells are especially abundant in the head and proximal portion of the optic nerve, which is involved in the early stages of axon regeneration. In addition, there is a significant increase in a heterogeneous population of Pax6+ cells in these brain regions after damage to the optic nerve [128]. Some of these cells have a neuroepithelial phenotype and constitute reactive neurogenic niches located in the periventricular and parenchymal zones of the brain.

Another population of Pax6+ cells has the RG phenotype and arises as a result of the activation of constitutive neurogenic domains, as well as during the formation of newly reactive neurogenic niches [127]. Therefore, following injury to the optic nerve, there is a pronounced neurogenic response in brain regions with direct retinal projections and in brain regions without such projections, as well as in remote areas, due to the activation of neurogenesis and the emergence of reactive neurogenic niches that contain NE cells and radial glia cells. These findings demonstrate that damage to the optic nerve induces increased reactive neurogenesis within the brain of adult trout.

Nevertheless, the study of the expression of Pax2 in the telencephalon of juvenile chum salmon *O. keta* showed that, as a result of acute TBI, there was a decrease in Pax2 expression. It was observed that mechanical brain injury increases the production of cystathionine beta-synthase (CBS) and glutamine synthetase in cells, but reduces the expression of Pax2 in juvenile chum salmon [14]. As a result of the injury, there was a significant decrease in the number of Pax2-expressing aNSCPs in the dorsal and medial pallium zones, as well as in the lateral zone of the subpallium.

To date, there is very limited data on the presence of adenovirus receptors in fish. The use of mouse-derived recombinant adeno-associated viral vectors (AAV) with a recently developed calcium indicator, GCaMP6s, has shown that these vectors can integrate into the neurons in the brain of juveniles of the Pacific chum salmon *O. keta* [129]. The gene was introduced in vivo by injecting a vector containing the GCaMP6–GFP (green fluorescent protein) directly into the mesencephalic region of a one-year-old chum salmon via an intracranial route. One week after a single vector injection, the insertion of AAV in the brains of juvenile chums was evaluated. AAV was expressed in various regions of the thalamus, the pretectum, the postero-lateral area, the postcommissural area, the medial and lateral areas of the diencephalon. The mesencephalic reticular formation of juvenile *O.keta* was assessed using CLSM 780, followed by immunohistochemical analysis to localize the early neuronal differentiation marker HuCD in conjunction with DAPI nuclear staining. The analysis results showed partial colocalization of GCaMP6m–GFP-expressing cells with the red fluorescent protein HuCD. Cells from the thalamus, posterior tuberal region, mesencephalic tegmentum, auxiliary visual system cells, mesencephalic reticular formation, hypothalamus, and postcomissural mesencephalic region of juvenile chum salmon that express GCaMP6m–GFP were assigned to the neuron-specific lineage of chum salmon brain cells, which indicates the ability of mammalian hippocampal rAAV to integrate into fish central nervous system neurons with subsequent viral protein expression. This is evident, as it indicates neuronal expression of mammalian adenovirus receptor homologue in juvenile chum salmon neurons.

## 7. Glial Plasticity in Response to Disease and Injury

One interesting aspect of zebrafish glia cells is their response to plasticity after disease or injury [106,130,131]. Different signaling pathways, such as Wnt and Shh, are involved in the regeneration of the optic nerve in zebrafish [132]. An analysis of the gene expression levels associated with these pathways has shown that the expression of the Wnt antagonist, Dkk1B, significantly decreases after injury. It is assumed that the use of a Wnt inhibitor, IWR1, reduces proliferation and differentiation in RG cells after injury, suggesting that upregulation of Wnt signaling is necessary for regeneration of the visual system. The Wnt/β-catenin and Notch signaling pathways are involved in NSCP differentiation and regulation of neurogenesis [130,133]. Studies have also shown that Shh signaling is activated specifically in RG cells after injuries, promoting proliferation and inhibiting differentiation into neurons. These results suggest that, in zebrafish, a high level of Notch signaling promotes RG at rest, and that an adequate level of Notch signaling is essential for the development of neonatal neurons from the RG. Under normal conditions, stimulation of Shh signaling or inhibition of Notch signaling also induces RG proliferation (Figure 5).

In fish, Sox2 and NeuroD are two of the key transcription factors involved in neurogenesis (Figure 5). Zebrafish, in particular, exhibit high levels of these factors during this process [134,135].

Histone deacetylases (HDACs) are one of the possible regulators of regeneration of the optic tectum zebrafish. Recent studies have shown that, in the damaged zebrafish tectum, the expression of HDAC1 and HDAC3 is significantly reduced. In order to analyze the role of HDACs in proliferation and differentiation of retinal ganglion cells (RG) after injury, pharmacological experiments were conducted using the HDAC inhibitor, trichostatin A. The results of these studies suggest that inhibition of HDACs after stab wounds suppresses RG proliferation, but promotes neuron differentiation [136]. Additionally, HDAC inhibition suppresses the reduction in the expression of target genes that transmit Notch, Her4.1, and Her6 signals after injury. These findings suggest that HDAC regulates regenerative neurogenesis by altering the expression of Notch target genes through deacetylation of histones. Histone acetylation and HDAC (histone deacetylase) are promising molecular targets for neural regeneration. Further research is needed to understand the molecular mechanisms behind the regulation of neural regeneration by histone acetylation.

Mitochondrial pathways also play an important role in neurogenesis in fish. The activation levels of mitochondrial factors, such as PGC-1α, can affect the energy metabolism of neural stem cells and their ability to differentiate [137].

NSCs in the adult zebrafish brain can also change their mode of division from asymmetric to symmetric after injury [38]. In essence, tight regulation of the quiescence-proliferation balance is a determinant of regenerative activity in the adult zebrafish brain and contributes to the maintenance of efficient stem cell pools that are ready to respond to neuronal loss through neuroregeneration. In addition, pathways involved in cell cycle regulation, such as cyclin-dependent kinases (CDKs) and cyclin-dependent kinase inhibitors (CDIs), are closely linked to the process of neuronal differentiation [138].

RGCs in the adult zebrafish brain also respond to neuronal loss by increasing the NSC proliferation and neurogenesis [106,107]. In the adult zebrafish model of induced Alzheimer’s disease, an injection of human amyloid-beta42 into the cerebrospinal fluid (CSF) was shown to form beta-sheet aggregates and lead to inflammation, synaptic degeneration, neuronal death, and cognitive decline [106,107]. Of particular interest is that zebrafish RGCs respond by inducing proliferative and neurogenic capacity, which is opposite to that of mammalian glia in Alzheimer’s disease. New neurons are formed and integrated into the circuit despite the prevailing amyloid toxicity. This regenerative capacity is mediated by neuronal expression of interleukin-4 (IL4), an anti-inflammatory factor that directly activates a subpopulation of radial glial cells through the il4r receptor and intracellular STAT6 signaling [139] or indirectly through a network of serotonergic neurons, periventricular neurons producing brain-derived neurotrophic factor (BDNF), and nerve growth factor receptor (NGFR)-positive neural stem cells [107]. The neurogenic response to beta-amyloid-42 is persistent but its magnitude decreases with age [140,141]. These mechanisms indicate a delicate balance between homeostatic proliferation and pathology-induced neurogenesis in the adult zebrafish brain.

It was initially surprising that, in zebrafish, the immune response seen after traumatic brain injury was actually necessary to trigger a regenerative process [142]. When the immune system is suppressed with drugs, the proliferation of RGCs and the subsequent regeneration of neurons are significantly reduced. On the other hand, sterile inflammation induced by injecting of the zymosan yeast particles into the ventricles of the brain stimulates proliferation of RGCs and generation of new neurons, even in the absence of injury [142]. 

Interestingly, similar findings were obtained in a study using a mouse model of optic nerve injury. Stimulation of the immune system, in addition to the injury, led to increased axonal regeneration. When using intraocular injections of zymosan, the authors observed that macrophages, neutrophils, and retinal microglia, all originating from monocytes, were activated. This activation was dependent on the dectin-1 receptor. As a result, axon regeneration increased in a specific manner. The authors hypothesize that this additional activation of immune cells results in secretion of inflammatory mediators that promote axon regeneration. They also suggest that immunosuppression can disrupt axon regeneration after spinal cord injury [143]. This disruption was observed in the larvae of zebrafish after treatment with immunosuppressive drugs [144].

## 8. Heterogeneity of Stem Cell Pools and Radial Glia

It has long been suggested that stem cell pools in vertebrates represent a homogeneous mass of cells giving rise to different cell types within this tissue [145]. Over the past two decades, several studies of zebrafish and axolotls have shown that stem cell pools can be heterogeneous and contain progenitors of specific cell types [146]. *Danio rerio* is a promising research model because the brain damage or complete cross-sections of the spinal cord are followed by effective neuroregeneration that successfully restores motor function. In the brain and spinal cord of zebrafish, stem cell activity is always able to reactivate the molecular programs needed for the regeneration of the central nervous system [147,148]. In mammals, traumatic brain injuries are followed by a decrease in neurogenesis and a poor ability for axon regeneration, which often results in insufficient functional restoration of the nervous tissue. Spinal injuries do not fully recover. The environment surrounding the stem cell niche formed by connective tissue and stimulating factors, including molecules that promote inflammation, seems to be a critical factor in triggering the stem cell proliferation or transdifferentiation into connective elements [148]. The study and comparison of neuronal regeneration in zebrafish and mammals may lead to a better understanding of mechanisms behind neurogenesis and the lack of a regenerative response in mammals. In particular, the role of inflammation is considered to be the main inhibitor of neuronal regeneration, and this can be investigated through the study of zebrafish.

Recent studies have shown that, in zebrafish, inflammation is a crucial process necessary for initiating regeneration [13]. These findings contradict many previous studies of mammals, in which the central nervous system was considered to be an immune organ and inflammation was thought to be one of the main negative factors leading to insufficient neuron regeneration [147]. At present, similarities and differences between vertebrates with natural regeneration abilities and those with limited or no regeneration capacity have been revealed. In particular, numerous factors that can actively modulate neurogenesis in adults have already been identified. For example, it has been found that stress and ageing negatively impact neurogenesis, while environmental enrichment and exercise may increase the formation of new neurons in the adult hippocampus.

To date, the phenomenon of adult neurogenesis has been widely studied in many different species and is considered an evolutionarily conservative trait [149]. Unlike mammals, teleost fish, such as the three-spined stickleback, gymnotiform fish *Apteronotus* sp. and zebrafish, have abundant sources of neurogenesis [29]. In these species, numerous neurogenic niches are distributed along the entire rostro-caudal axis of the brain, where continuous turnover of neurons is found up to adulthood [29]. It is noteworthy that, in addition to constitutive neurogenesis throughout life, teleosts are also considered as powerful regenerators after traumatic injury [58,149,150].

In order to create a knife wound in the telencephalon of an adult zebrafish, a small cannula is inserted through the nose into the brain parenchyma, without directly affecting the area of the ventricles where NSCs are located [58]. In the case of injury, there is a rapid cellular response in the form of increased apoptosis and edema formation.

RGSc respond to injury by increasing their proliferation in the damaged hemispherical ventricle zone, and experiments that track genetic lines confirm that the daughter cells of these cells differentiate into new neurons that repopulate the injured area [58]. In addition, there is an increase in the level of GFAP and hypertrophy of the glial processes, indicating reactive gliosis. However, neither chronic inflammation nor scarring occurs in adult zebrafish brains after injury.

So, we will discuss NSCs and progenitor cells from different species and how they respond to acute damage to the central nervous system. Then, we will explore how different organisms respond to injury by activating their immune systems. We will also discuss important types of immune cells, with specific reference to their effects on the behavior of neural stem cells. Finally, we will provide an overview of the main inflammatory mediators released during injury that have been linked to neural stem cell activation and regeneration.

In the mammalian CNS, neural stem cells are heterogeneous [110,111]. To study traumatic brain injuries in rodents, various injury paradigms have been established, such as models of weight loss, fluid shocks, or cortical puncture wounds [151]. Mammals exhibit the lowest regenerative capacity, whereas non-mammalian vertebrates such as chicks, amphibians, and fish exhibit higher potential [13]. Similar to areas of continuous proliferation in the adult brain, the ciliary marginal zone (CMZ) of birds, amphibians and fish functions as a niche of stem cells, contributing to both adult retinal growth and its regeneration [152]. In the retina of amphibians, neuron progenitor cells are generated by transdifferentiation of pigmented epithelial cells [13,153]. During the homeostasis of the Muller glia, the danio fish gives rise only to rod-shaped photoreceptors [154]. This changes when the retina is damaged: Muller’s glia partially dedifferentiates, undergoes interkinetic migration of nuclei, re-enters the cell cycle and forms one neuronal precursor cell by asymmetric cell division. This precursor undergoes subsequent cell divisions and forms a neurogenic cluster that migrates to the site of damage, where cells differentiate into lost types of neurons [155]. During regeneration, various signals involved in the response to stress, inflammation, gliosis and cell adhesion are necessary and sufficient to stimulate the proliferation of Muller glia [13].

In the zebrafish nervous system, several markers have been used to label radial glial cells or astroglia (e.g., GFAP, S100b, her4.1, nestin). Neurogenic capacity has been inferred based on proliferative status (labeled by PCNA or Mcm5) or predominant Notch signaling [106,107]. Reactive proliferation has also been observed after spinal cord injury in zebrafish, when ependymal radial glial cells begin to proliferate and generate new motor neurons [148]. Newly created neurons mature and show signs of terminal differentiation and integration into the circuit [155]. Radial glia migrates to the severed region of the spinal cord, where it forms a “glial bridge” that supports the regeneration of axons at the site of injury [156,157]. After the lesion, a large number of interneurons of different types are also detected. However, they are not formed from Olig2+ progenitor cells, but from another Pax6+ Nkx6.1+ precursor domain. New V2 interneurons are generated in the domain of the ependymal layer, which are usually absent in the unaffected spinal cord [158]. Despite the fact that altered serotonergic and dopaminergic innervation is observed in danio fish, locomotor function is restored after 6 weeks, which indicates a high plasticity of the spinal network of adult individuals [159].

Advances in technology, especially single-cell transcriptomics, have now opened up a new way to study the heterogeneity of RGCs. This approach has identified different types of neural progenitor cells after sorting her4.1-positive glial progenitors [106,160]. Pre- and proliferating neuroblasts, two identified types of neural progenitors, contain very different molecular programs, suggesting that neuroblasts can significantly change their molecular profiles towards the cell lineage they will differentiate. Indeed, some neuroblast markers can be seen in early neuron types, indicating a continuum between specific neuroblasts and their lineages [106]. Other progenitor cell clusters have provided further evidence that the adult zebrafish brain contains heterogeneous progenitors that respond differently to different environmental cues or insults. In the zebrafish model of brain amyloid toxicity, only two types of progenitor cells located in the dorsomedial region of the adult zebrafish telencephalon increased their proliferation [106,107]. Together with the highly divergent molecular programs of all types of neural progenitor cells in the zebrafish telencephalon, this demonstrated cell heterogeneity in the neural stem cell pool. Some progenitors express genes associated with astroglial functions (e.g., aldh1l1, aldoc, ndrg2) and others express ependymal markers (e.g., cilia components), thus, indicating that either some glial cells are deposited as quiescent and non-neurogenic or these cells, coexisting with neurogenic precursors, represent a defined physiological state and can become neurogenic under certain conditions [106].

Another recent single-cell transcriptomics study considered the transcriptional profiles of early newborn neurons [160] and confirmed their heterogeneity, as was observed in previous studies [106,139]. Finally, lineage tracing experiments revealed that heterogeneous adult neural stem cells are hierarchically organized into deeply quiescent and self-renewing reservoir stem cells that support constitutive neurogenesis in zebrafish throughout life [107]. The fast-growing teleost fish *Nothobranchus furzerii* (killifish) has recently been shown to contain non-glial progenitors that divide rapidly and promote rapid brain growth [161]. It has been suggested that non-glial progenitors delay their transition to a resting state and may, therefore, contribute to the rapid growth required for short-lived killifish [161]. This heterochrony of neuronal progenitors may be a major determinant of the neurogenic and regenerative properties of bone marrow and may play an important role in understanding the balance between quiescence and proliferation in health and disease. Therefore, understanding the mechanisms of efficient nerve regeneration in zebrafish may inspire the development of new treatments for various neurodegenerative diseases.

In the adult red-spotted newt *Notophthalmus viridescens*, the neurotoxin-mediated damage model selectively removes dopaminergic neurons, causing a strong inflammatory reaction with recruitment and activation of microglia. The salamander completely regenerates these lesions by activating ependymoglium-like RG cells that proliferate and regenerate lost dopaminergic neurons [162]. Interestingly, in immunosuppression using dexamethasone, more new tyrosine hydroxylase neurons are detected in response to damage [163]. The analysis of the immune response in the temporal aspect is also extremely important, since the correct time is probably one of the key factors influencing the regenerative result. In addition, it will be extremely important to continue studying the basic cellular and molecular inflammatory signals that act on neural stem cells.

## 9. The Role of Astroglia in the Functioning of Neural Circuits and Animal Behavior

Astroglia are a type of glial cell in the central nervous system (CNS) of vertebrates, including humans. These cells play a crucial role in the development and function of the nervous system, particularly in the process of axon myelinization, maintaining cerebrospinal fluid homeostasis, and regulating the ion exchange in neurons [164]. 

In adults, astrocytes continue to perform their functions. They are involved in the regeneration of axons and the recovery process after injuries or diseases, such as stroke and Alzheimer’s disease. Additionally, they can regulate synaptic plasticity and adapt to learning and memory. Astroglia functions may change with age. For example, during aging, neuronal loss and decreased myelination occur, what can lead to deterioration of astroglia functions. However, studies have shown that astroglia is able to partially compensate for these changes and maintain the normal functioning of the nervous system throughout life [165].

In general, astroglia play a critical role in formation, function, and regeneration of the nervous system. Understanding the evolutionary changes that occur in these cells can contribute to the development of new treatment methods and preventive strategies for CNS diseases [166].

Recent evidence highlights the important role of astroglia in regulating of neural activity, brain states, and animal behavior in both vertebrates [167,168] and invertebrates [169]. Studies on rodents have shown that astroglial cells are highly dynamic brain components that respond to locomotion [170] or sensory stimulation [171] with pronounced changes in astroglial calcium levels and may regulate learning [172,173] or other state transitions [174,175] in the brain.

Some studies have described the effect of astrocytes on neuronal activity and behavior [172,176]. However, we believe that there are further prospects in this field that could help to clarify the mechanisms behind behavioral processes. In the near future, we hope that conventional models and non-model organisms will provide answers to questions about how astrocytes detect neuronal activity or environmental changes and what signaling pathways are involved in their responses.

Norepinephrine [175] and acetylcholine [177] are assumed to be the main triggers of astroglial activation, but several other molecules, including glutamate [178], play important roles in astroglial physiology. Consequently, activation of astroglia triggers multiple cellular processes leading to the release of gliotransmitters such as glutamate [179], adenosine triphosphate (ATP) [180], D-serine [181], and GABA A [182], which, in turn, alter neuronal activity. It has been suggested that zebrafish brain astroglia cells are a homologue of mammalian astrocytes [104,183]. Despite the morphological differences, an important finding that shows a similarity between zebrafish and mammalian astroglia cells is the presence of orthologs of the glutamate transporter EAAT2 in retinal Müller glial cells [184].

This study has also demonstrated that glutamate transporters alter the dynamics of glutamate-mediated neuronal activity, which is consistent with the important role that this transporter performs in mammals by clearing excess glutamate through astroglia [185]. In fact, mice lacking the glutamate transporter EAAT2 exhibit epileptic seizures due to excess glutamate [186]. Similarly, disruption of astroglial–neuron interactions during epileptic seizures in zebrafish coincides with excessive glutamate release [187].

We believe that further applications of optogenetic techniques and vital neuroimaging in studying of neuroglial interactions in regenerating models, particularly in fish, will further elucidate and enhance our current understanding of the diverse functions of RGCs [167]. Since, in fish, radial hindbrain cells (RG cells) are found in the adult brain and, as recent studies have demonstrated [168], they are not restricted to structural and neurogenic roles, but actively participate in modulating neuronal network activity. This provides new avenues for exploring RG cell properties in vertebrates. 

Initial steps in this direction have been made through a study of juvenile Pacific salmon. In the work related to the long-term monitoring of connections between the cerebellum and other parts of the brain in juvenile chum salmon, researchers have found that recombinant adeno-associated virus (rAAV) vectors can integrate into the neurons of various regions of the brain after being injected into the cerebellum [188]. The distribution of recombinant adeno-associated viral vectors (rAAVs) after the injection of the base vector into the cerebellar tissue of juvenile chum salmon showed highly specific patterns of gene expression in bipolar neurons in the laterocaudal lobes of the tectum opticum. The rAAVs were detected in the dorsal thalamus, the epithalamus, the round nucleus, and the pretectal complex, indicating a targeted distribution of the gene through thalamocerebellar connections. GFP expression was detected in cells of the epiphysis and the posterior tubercles of juvenile chum salmon, which is associated with the distribution of the rAAV and the flow of cerebrospinal fluid through the ventricles of the brain and its outer surface. Direct delivery of rAAV to the central nervous system via intracerebral injection allows it to distribute widely throughout the brain. This is evident in the presence of specialized projection areas in juvenile salmon’s cerebellum, as well as outside of it, and in the identification of transgene expression within these areas. These findings confirm the potential for rAAV to spread through both intracerebral and extracerebral projections of the cerebellum in both the lateral and medial pallium zones.

Mammalian astroglia cells form an extensive functional syncytium through gap junctions that can rapidly redistribute ions, neurotransmitters, and other molecules, such as ATP, over long distances in the brain [189]. In fact, this tightly coupled network is so efficient in redistributing ions and neurotransmitters that mice lacking the astrocytic gap junction connexin 30 and connexin 43 exhibit spontaneous seizures [190]. Similarly, zebrafish astroglial networks have widespread gap junction connectivity and express connexin43 [187]. Mammalian astroglial gap junction connectivity has been shown to mediate spatiotemporal synchrony in the activity of neural and astroglial networks [191]. Similarly, zebrafish astroglia cells exhibit highly synchronous activity during different neuronal states [187]. Further studies are needed to better understand the function of gap junction connectivity in the astroglial networks of zebrafish and mammals.

Thus, the discussion of slot connections is particularly interesting. It is necessary to further investigate how gap junctions influence the coordination of astroglial and neural networks in different brain conditions or in response to injury. Another intriguing and significant area is the exploration of the potential for targeting of discontinuous junctions in therapeutic strategies for the management of neurological disorders.

The expansion of our knowledge on functional consequences of astroglia-neuron interactions in both physiological and pathophysiological conditions represents a promising direction for future research. To advance this trend, it is crucial to develop new high-tech techniques for vital neuroimaging, as well as to employ experimental approaches on both relatively well-studied laboratory mammals and more complex and less-studied models.

Thus, in molecular terms, zebrafish RGCs are similar to mammalian astrocytes, and can therefore also be referred to as astroglia. However, a question exists as to if zebrafish astroglia cells interact functionally with neurons, in a similar way to their mammalian counterparts. In fact, a recent study has clearly demonstrated that astroglia in the hindbrain of larval zebrafish are activated by norepinephrine [168,192], in a similar way to mammalian astrocytes [175]. These results indicate a highly conserved mechanism for the recruitment of astroglial networks in vertebrates. Moreover, the same study [192] has also shown that activation of hindbrain astroglia leads, in several different ways, to a transient cessation of neuronal activity, which is assumed to be mediated by astroglial activation of GABA–ergic neurons. In parallel, another recent study [187] has shown that activation of forebrain astroglia by channelrhodopsin2 results in marked excitation of nearby neurons, mediated by glutamate and gap junctions. These results indicate that, as in mammals, the zebrafish astroglial and neuronal networks directly interact with each other via distinct pathways.

These results highlight the pronounced astroglial–neuron interactions in both zebrafish and mammals. The question of if the function of astroglia–neuron communication is conserved across vertebrates remains to be answered. However, two recent studies of larval zebrafish point to clear parallels with mammals, where astroglial–neuron interactions play a direct role in various state transitions of neural networks during both physiological and pathophysiological conditions. One of these studies [168] has shown a clear role of astroglial networks in accumulating input and altering neural activity in ways that suppress failed swimming attempts. Another study [187] showed that interactions between astroglia and neurons are not stationary, but highly dynamic, and also that disruption of these interactions led to the propagation of epileptic seizures. Combined with previous demonstrations of the functional roles of other types of glia in zebrafish [193,194], all of these studies can most likely be considered as a prelude to subsequent studies that will further clarify the mechanisms and functions of astroglial–neuronal interactions in the zebrafish brain. As in mammalian astrocytes, these interactions are likely to be mediated by multiple processes and molecules that act in parallel and, in turn, possibly play multiple roles. Because of the transparency, small size, and optogenetic ability of zebrafish, it is evident that future studies of this fish will complement those of rodents and provide important information about the cellular and physiological processes that underlie astroglial function and modulate neural activity and behavior in animals.

## 10. Conclusions

Despite the meaningful results obtained, it is necessary in the future to focus research on identifying neurogenic areas in other regions of the brain in Pacific salmon, particularly in the *medulla oblongata* and spinal cord, as well as in the hypothalamic-pituitary system. Additionally, it will be important to investigate mechanisms for enhancement of proliferative activity in these areas after TBI. Several issues related to the study of ultrastructural and immunohistochemical organization of secondary matrix zones within the brain parenchyma and on the surface of the brain, as well as different sensory systems within the CNS, and the characteristics of NSCPs during ontogenesis, should be addressed to determine the mechanisms of neuronal plasticity after injury. 

Studying the mechanisms involved in neuroprotection through the action of transcription factors, which are involved in neurogenesis, is crucial for understanding of the brain recovery after TBI [195]. The activation of the brain NSCs in fish is linked to changes in expression of certain molecular markers, such as Sox2 and NeuroD [134,135]. These molecular markers are regulated by other transcription factors, which can also affect the regenerative capacity of the fish brain. The use of pharmacological agents that target these molecular markers and transcription factors can therefore enhance the regenerative abilities of the brain in fish.

Compared to mammals, in which CNS regeneration is limited, zebrafish have demonstrated the ability for extensive regeneration even of the adult nerve tissues. This raises interest in the mechanisms underlying this difference. 

The complete doubling of the genome in Salmonidae is probably one of the reasons for the presence and long-term preservation of NE cells in their CNS and their slow replacement by RG cells (representing aNSCPs [58]) and absence of these cells in the postembryonic development in mammals. Transcription factors, signaling and cellular pathways represent a complex network of interacting elements that regulate neurogenesis in fish. The study of these molecular mechanisms opens up new prospects for the development of strategies to stimulate neurogenesis and, possibly, the treatment of neurological disorders in humans. Thus, therapeutic strategies should probably focus on the possibility of activating these zones, searching for inhibitors or activators of neuroprotection factors and the Wnt and Notch signaling pathways regulating neurogenesis [130,132]. We believe that in future studies it is appropriate to consider the following hypotheses: 

(i).How can astroglia contribute to the resistance or vulnerability of neural circuits to injury or disease?(ii).What are the potential therapeutic targets in astroglial signaling pathways? 

Going forward, a deeper dive into how astroglia functions adapt to specific physiological needs or environmental challenges faced by different species can provide fascinating insights into evolution. In particular, the answer to the question looks intriguing, how can astroglia’s response to neurotransmitters such as norepinephrine or acetylcholine differ in species with completely different lifestyles or habitats? This area can become very promising and useful in future areas of research on the relationship between neurons and glia. The results of this research may lead to new insights. By combining the already known functional relationships between astrocytes and neurons, with the discovery of new and intriguing aspects of the interaction between nerves and glial cells, we can gain a better understanding of how astrocyte-neuron interactions contribute to the regulation of complex behavior and the development of neurological conditions. This research may also highlight the clinical significance of such findings, which could lead to the development of new treatments for neurological disorders.

## Figures and Tables

**Figure 1 ijms-25-03658-f001:**
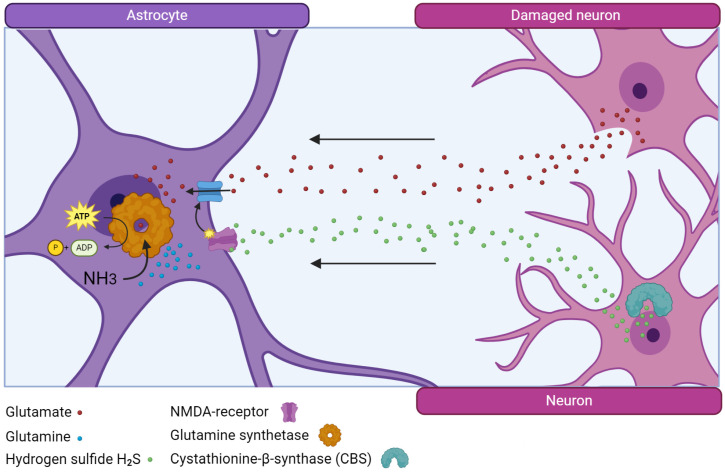
Glutamate utilization and increased hydrogen sulfide (H_2_S) production after acute traumatic brain injury (TBI) of juvenile chum salmon, Oncorhynchus keta. At the top right, a damaged glutamatergic neuron is shown excreting glutamate into the intercellular environment, which is captured by the astrocyte through NMDA receptors and then converted into neutral glutamine through GS. The intact neuron at the bottom right responds to TBI by increasing the production of hydrogen sulfide, which diffuses freely into the intercellular space, exerting a neuroprotective effect on the damaged neuron.

**Figure 2 ijms-25-03658-f002:**
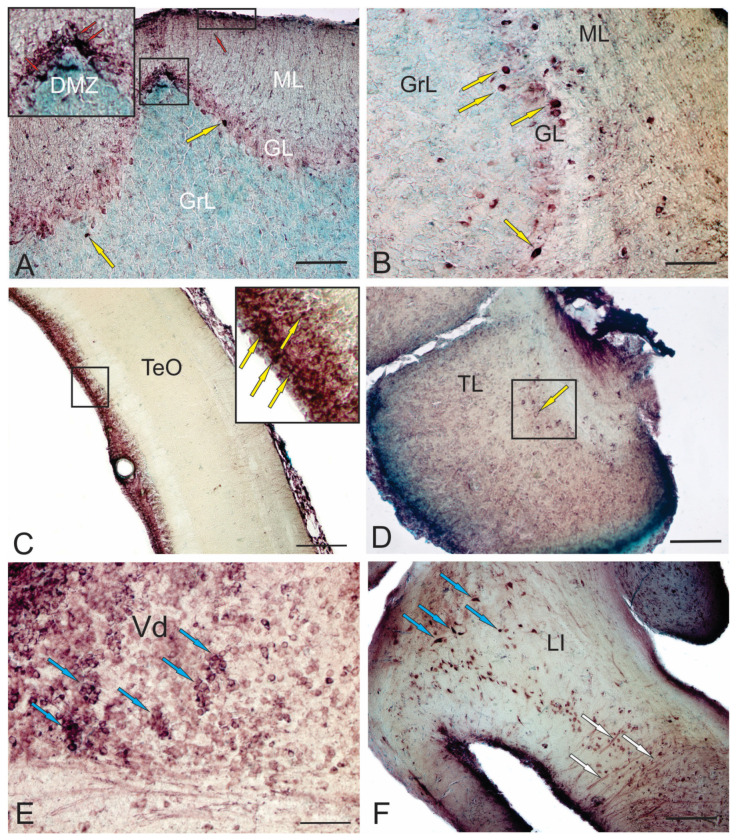
Immunohistochemical labeling of doublecortin and transcription factor Pax6 in the brain of adult trout *Oncorhynchus mikkis*. (**A**)—DCX+ cells (yellow arrows) in the ganglion layer (GL) of the cerebellum, in the inset dorsal matrix zone (DMZ), containing DCX+ cells (red arrows). From above, in a black rectangle, DCX+ cells in the molecular layer (ML). (**B**)—DCX+ projection neurons (Purkinje cells and Eurydendroid cells) in the basal part of the cerebellum (yellow arrows), GrL—granular layer. (**C**)—DCX+ neurons in the optic tectum (an enlarged fragment is shown in the black inset). (**D**)—DCX+ cells in the *torus longitudinalis* (in the black square, TL). (**E**)—DCX+ cells (blue arrows) in the Vd of subpallium. (**F**)—Pax+ cells (blue arrows) and radial glia (white arrows) in the *lobus inferior* (LI) of the hypothalamus. Scale bar: (**A**,**B**,**D**)—200 mm, (**C**,**F**)—100 mkm, (**E**)—50 mkm.

**Figure 3 ijms-25-03658-f003:**
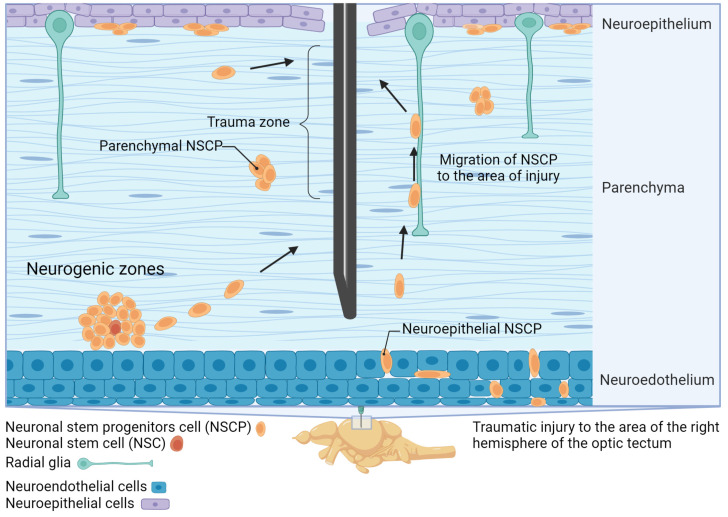
Acute traumatic brain injury in juvenile chum salmon *O. keta*. A needle was used to create a TBI to the right hemisphere of the optic tectum. As a result of TBI, the processes of proliferation of NSCPs localized in the primary periventricular neurogenic zones containing resident NSCs, as well as areas of secondary neurogenesis (reactive neurogenic niches), were activated. The newly formed cells migrated to the area of injury along the processes of radial glia cells.

**Figure 4 ijms-25-03658-f004:**
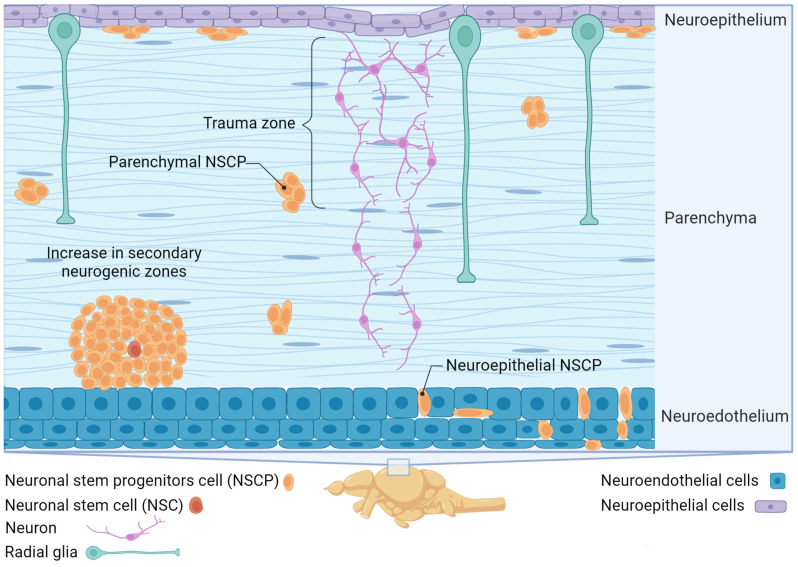
Regenerative response to long-term brain damage in juvenile chum salmon *O. keta* at 3 months after TBI. The injury area is filled by new neurons, and the number and size of areas of secondary neurogenesis (reactive neurogenic niches) increases.

**Figure 5 ijms-25-03658-f005:**
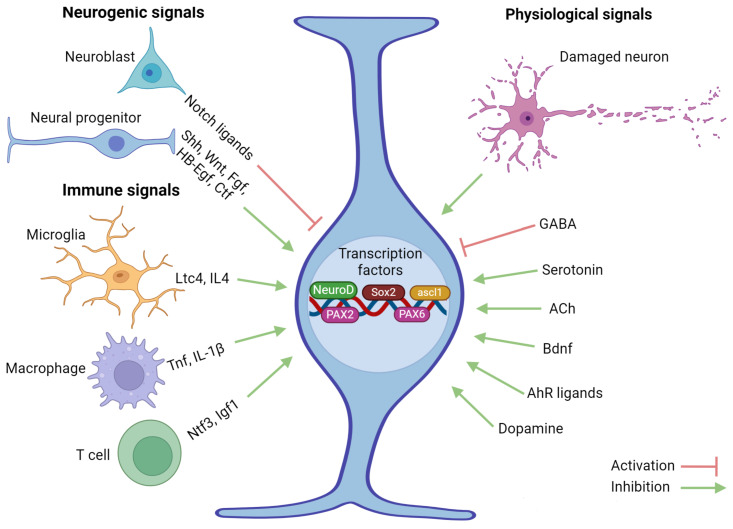
Factors affecting neurogenesis in the fish brain. The diagram illustrates various factors that influence neurogenesis, including immune signals (on the **left**), which come from glial cells of different types (microglia, macrophages, and T-like cells). These signals initiate molecular cascades, such as Notch, Shh, and Wnt. Physiological signals (on the **right**), such as neurotransmitters like GABA, serotonin, Ach, and dopamine, and neuroactive substances like Bdnf, also play a role in regulating neurogenesis. All of these external factors activate different transcription factors in the nucleus and regulate the process of neurogenesis by activating or inhibiting it.

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
