# Peer review of "Adult Neurogenesis of Teleost Fish Determines High Neuronal Plasticity and Regeneration"

_ijms, 2024, doi:10.3390/ijms25073658_

Round 1
Reviewer 1 Report
Comments and Suggestions for Authors
This review article titled "Adult neurogenesis of teleost fish determines high neuronal plasticity and regeneration" presents an in-depth examination of the remarkable capabilities of teleost fish for adult neurogenesis and its implications for understanding neural plasticity and regeneration. Teleosts, with their unparalleled capacity for brain repair and growth, serve as excellent models for exploring the mechanisms of neurogenesis beyond the limited scope observed in mammals. The manuscript emphasizes the crucial roles of neural stem and progenitor cells (NSPCs) located in specific brain regions and the molecular pathways that guide their proliferation, differentiation, and integration into neural circuits. It contrasts these processes with those in mammalian systems, highlighting the significant potential for regenerative strategies. The review also explores the functional outcomes of such plasticity, including adaptation to environmental changes and recovery from injury, suggesting that teleost fish offer valuable insights for developing new approaches to treat neurodegenerative diseases and brain injuries in humans. Through comparative analysis, this manuscript enriches our understanding of neuronal regeneration, suggesting a reconsideration of the regenerative capabilities of the adult brain across species.
To enhance the quality of your manuscript, we recommend the following improvements:
1. The manuscript is well-structured, dividing the discussion into sections that first address general aspects of CNS regeneration across vertebrates before delving into the specific case of Pacific salmon. However, it could benefit from a clearer introduction to the significance of studying different vertebrate models, particularly emphasizing why the Pacific salmon model offers unique insights compared to other models.
2. The manuscript references a wide range of studies, indicating a thorough literature review. Nonetheless, it might be beneficial to include more recent studies that might have been published on the topic, ensuring the review's comprehensiveness and up-to-dateness. Furthermore, a more detailed comparison with other notable models of CNS regeneration, such as the axolotl or certain lizard species, could enrich the discussion.
3. While the manuscript focuses on the results and implications of NSPCs' regenerative capabilities, it lacks specific details about the methodologies used in the referenced studies. Including brief descriptions of the experimental designs, such as the types of injuries induced or the genetic markers used to identify NSPCs, could provide readers with a better understanding of how conclusions were drawn.
4. The interpretation of the findings, particularly regarding the unique properties of salmon NSPCs and their implications for CNS regeneration, is insightful. However, the manuscript could further discuss the limitations of the current understanding or conflicting findings in the field. This would offer a more balanced view and highlight areas requiring further investigation.
5. The discussion on therapeutic prospects is promising but somewhat speculative. Expanding on how the findings from salmon and other vertebrates could be translated into mammalian models, including humans, would make this section more compelling. This could include potential challenges, necessary technological advancements, or ethical considerations.
6. The manuscript concludes with a strong statement on the significance of the findings but could benefit from a clearer outline of specific future research directions. Identifying key questions that remain unanswered and proposing potential experimental approaches to address them would be valuable.
7. Ensure that all scientific terms are used correctly, and abbreviations are defined upon first use. Also, double-check references for accuracy and completeness, ensuring that all cited works are correctly attributed and accessible.
8. Lastly, integrating the findings into the broader context of CNS regeneration research could enhance the manuscript. Discussing how the insights from salmon NSPCs contribute to the overall understanding of regeneration across species would underscore the study's relevance.
9. The manuscript does an excellent job of contrasting the regenerative capabilities of aNSPCs in fish (specifically zebrafish) and mammals. However, it could benefit from a deeper comparative analysis on how these differences in regenerative capacities might inform therapeutic strategies. Specifically, identifying key factors that enable zebrafish's high regenerative potential could offer insights into overcoming limitations in mammalian CNS regeneration.
10. While discussing the behavior of aNSPCs and their role in regeneration, the manuscript could provide more details on the methodologies used to study these cells in both zebrafish and mammals. This includes specifics on intravital imaging, lineage analysis, and other experimental approaches that yield insights into aNSPC behavior, neurogenesis, and neuron integration into existing circuits.
11. The discussion on the molecular markers of aNSPCs and the signaling pathways involved in their activation and proliferation is informative. Expanding on these molecular mechanisms, particularly those that differ between regeneratively competent and incompetent species, could enrich the narrative. Highlighting recent discoveries or ongoing debates in this area would also be beneficial.
12. The potential use of aNSPCs in regenerative therapy is a crucial aspect of this discussion. The manuscript might further explore the current barriers to using these cells in mammalian brain repair, including issues related to cell differentiation, integration, and functional recovery post-injury. Discussing ongoing research or experimental therapies that aim to overcome these barriers could provide a hopeful perspective on future advancements.
13. The manuscript mentions neurogenic niches in both zebrafish and mammals, focusing on the SVZ and SGZ in mammals. Providing more detail on the characteristics, regulation, and functional significance of these niches in both species could offer readers a deeper understanding of the context in which aNSPCs operate. This could include discussions on the niche environment, cellular interactions, and how these factors influence neurogenesis.
14. While the manuscript effectively reviews the current state of knowledge, it could more explicitly outline areas where further research is needed. This may include unanswered questions about the molecular pathways governing aNSPC behavior, challenges in translating findings from zebrafish to mammals, or the exploration of novel neurogenic regions outside the traditional niches.
15. Ensure scientific terms are accurately used and fully explained where necessary. For example, when introducing molecular markers and signaling pathways, briefly describe their roles in neurogenesis to cater to readers who may not be familiar with these terms. Additionally, verifying that all references are current and relevant would strengthen the manuscript's credibility.
16. Including figures that visually summarize the differences in aNSPC characteristics, their localization in the brain, and their regenerative processes across species could enhance reader comprehension. Diagrams depicting the molecular pathways involved in aNSPC activation and differentiation could also be helpful.
17. The manuscript does a commendable job highlighting the unique neurogenic capabilities of Pacific salmon. Expanding the comparative analysis with more vertebrate models, particularly those with limited regenerative capabilities, could further underscore the significance of the findings and help elucidate potential evolutionary pathways or mechanisms that have led to these differences.
18. While the document discusses the activation of NSPCs and their migration to injury sites, a more detailed exploration of the underlying molecular and cellular mechanisms would be beneficial. Specifically, identifying signaling pathways, transcription factors, and environmental cues that govern these processes in Pacific salmon could provide deeper insights into the basis of their regenerative prowess.
19. The manuscript could benefit from additional information on the experimental approaches used to study NSPCs in Pacific salmon, including techniques for tracking cell proliferation, migration, and differentiation in vivo. Such details would help readers appreciate the complexity of the research and the validity of the conclusions drawn.
20. Discussing the functional integration of newly formed neurons into existing neural circuits and their impact on the behavior or physiological functions of Pacific salmon would enhance the manuscript. This aspect is crucial for understanding the biological significance of the observed neurogenic processes, particularly in the context of the salmon's life cycle and adaptation to environmental changes.
21. The manuscript hints at the potential implications of salmon NSPCs for regenerative medicine. Delving into specific strategies for translating these findings into mammalian models, including humans, could make the discussion more relevant to the broader field of neurodegenerative disease research and treatment development.
22. The mention of NSPCs in older individuals introduces an interesting aspect of age-related changes in neurogenic capacity. Expanding on how the properties of NSPCs and their regenerative potential change with age, and the factors influencing these changes, would add depth to the discussion on the plasticity and resilience of the salmon brain.
23. The manuscript would benefit from a clearer articulation of future research directions, particularly addressing the gaps in understanding the regulation of NSPCs, their long-term functional integration, and the possibility of harnessing these cells for therapeutic purposes. Highlighting specific questions and potential experimental approaches would guide future studies in this exciting area of research.
24. The comparison between zebrafish and mammals offers a valuable perspective on the evolutionary conservation of astroglial functions. To deepen this analysis, consider discussing the implications of these similarities and differences for our understanding of astroglia's roles across the animal kingdom. Highlighting specific evolutionary advantages or adaptations in astroglial functions could provide intriguing insights.
25. While the section effectively outlines the impact of astroglia on neural activity and behavior, it could benefit from more detailed explanations of the underlying mechanisms. For example, how do astroglial cells detect neuronal activity or environmental changes, and what are the signaling cascades involved in their response? Expanding on these processes would enrich the narrative.
26. The manuscript mentions the use of optogenetics and other methodologies to study astroglial-neuronal interactions. Delving into these techniques, their advantages, and limitations could offer readers a clearer understanding of how current findings were obtained and what future studies might look like. Discussing the potential of emerging technologies, such as in vivo imaging and single-cell transcriptomics, could also be valuable.
27. Expanding on the functional implications of astroglial-neuron interactions in both physiological and pathophysiological states would enhance the section. Specifically, how do these interactions contribute to the regulation of complex behaviors or the development of neurological disorders? Addressing these questions could highlight the clinical relevance of the research.
28. The discussion on gap junctions is particularly intriguing. Further elaboration on how gap junction connectivity influences the coordination of astroglial and neuronal networks during different brain states or in response to injury would be beneficial. Additionally, exploring the potential for targeting gap junctions in therapeutic strategies for neurological conditions could be interesting.
29. The section concludes with an optimistic outlook on future research using zebrafish as a model organism. Articulating specific questions or hypotheses that future studies should address would be helpful. For instance, how might astroglia contribute to the resilience or vulnerability of neural circuits to injury or disease? What are the potential therapeutic targets within astroglial signaling pathways?
30. A deeper dive into how astroglial functions are tailored to the specific physiological needs or environmental challenges faced by different species could offer fascinating evolutionary insights. For example, how might the astroglial response to neurotransmitters like norepinephrine or acetylcholine differ between species with vastly different lifestyles or habitats?
31. Briefly touching upon the role of astroglia in neurodevelopment and how their functions evolve from embryonic stages to adulthood could provide a more comprehensive view of their importance in the CNS.
Comments on the Quality of English LanguageEnsure that sentences are clear and to the point. Avoid overly complex or lengthy sentences that may obscure the intended message. Simplifying sentence structures can enhance readability and ensure that your findings are accessible to a broad audience.
Author Response
Many thanks to the respected reviewer for their very attentive, comprehensive, and detailed comments. We are sincerely grateful to accept these suggestions to improve the quality of our manuscript. Thank you for your great and extremely valuable work for us.
To enhance the quality of your manuscript, we recommend the following improvements:
- The manuscript is well-structured, dividing the discussion into sections that first address general aspects of CNS regeneration across vertebrates before delving into the specific case of Pacific salmon. However, it could benefit from a clearer introduction to the significance of studying different vertebrate models, particularly emphasizing why the Pacific salmon model offers unique insights compared to other models.
Thank You for Your recommendation. We have added information to the section "Biological features of NSPCs in Pacific salmon", explaining why the Pacific salmon model provides unique information when compared to other models.
- The manuscript references a wide range of studies, indicating a thorough literature review. Nonetheless, it might be beneficial to include more recent studies that might have been published on the topic, ensuring the review's comprehensiveness and up-to-dateness. Furthermore, a more detailed comparison with other notable models of CNS regeneration, such as the axolotl or certain lizard species, could enrich the discussion.
Thank You for this comment, according to the recommendations, more recent links have been added to the manuscript. The section "Heterogeneity of stem cell pools and radial glia" provides data on other groups of vertebrates.
- While the manuscript focuses on the results and implications of NSPCs' regenerative capabilities, it lacks specific details about the methodologies used in the referenced studies. Including brief descriptions of the experimental designs, such as the types of injuries induced or the genetic markers used to identify NSPCs, could provide readers with a better understanding of how conclusions were drawn.
Thank You for this recommendation, according to which some methodological and experimental schemes for the identification of NSPCs and neurogenic niches have been added to the section "NSC in the adult brain".
- The interpretation of the findings, particularly regarding the unique properties of salmon NSPCs and their implications for CNS regeneration, is insightful. However, the manuscript could further discuss the limitations of the current understanding or conflicting findings in the field. This would offer a more balanced view and highlight areas requiring further investigation.
Thank you for this request, an addendum has been inserted into the revised version of the manuscript.
- The discussion on therapeutic prospects is promising but somewhat speculative. Expanding on how the findings from salmon and other vertebrates could be translated into mammalian models, including humans, would make this section more compelling. This could include potential challenges, necessary technological advancements, or ethical considerations.
The following addendum has been inserted for this comment:
Despite the limited number of neurogenic zones in mammals, (Ming GL, Song H. Adult neurogenesis in the mammalian brain: significant answers and significant questions. Neuron. 2011;70 (4):687-702.), the mechanisms of neuroprotection remain conservative, for example, glutamine synthase is involved in converting glutamate into neutral glutamine and reducing exitotoxicity, as well as regulating the processes of neurogenesis, including the formation of new neurons and their introduction into existing neural networks in the hypocampus in mammals (Gao K, Wang G, Wang Y, Han D, Bi J, Yuan Y. Neurogenesis in the adult brain and its implications for Alzheimer's disease. J Cell Mol Med. 2018;22(7):3103-3110.) (Ming GL, Song H. Adult neurogenesis in the mammalian brain: significant answers and significant questions. Neuron. 2011;70(4):687-702). Aromatase catalyzes the conversion of testosterone to estrogen, playing a key role in the synthesis of estrogens that are involved in brain recovery after injury (Kim YS, Kim JJ, Kim JY, et al. Stereotaxic Infusion of 17β-Estradiol Enhances Recovery After Traumatic Brain Injury in Female Rats. Endocrinology. 2015;156(11):4351-4360.)
Hydrogen sulfide can act on a number of molecular targets, such as cytochrome C and caspases, to prevent programmed cell death. In addition, there is evidence that H2S can promote neurogenesis, which is also an important aspect of brain repair (Hu LF, Lu M, Tiong CX, Dawn GS, Hu G, Bian JS. Neuroprotective effects of hydrogen sulfide on Parkinson's disease rat models. Aging Cell. 2010;9(2):135-146)(Kimura H. Hydrogen sulfide as a neuromodulator. Mol Neurobiol. 2002;26(1):13-19.)
Thus, inhibitors or activators of neuroprotection factors may become targets for the development of new therapeutic strategies
- The manuscript concludes with a strong statement on the significance of the findings but could benefit from a clearer outline of specific future research directions. Identifying key questions that remain unanswered and proposing potential experimental approaches to address them would be valuable.
Thank you for your comment, we have added information in accordance with the recommendations in the Conclusion section.
- Ensure that all scientific terms are used correctly, and abbreviations are defined upon first use. Also, double-check references for accuracy and completeness, ensuring that all cited works are correctly attributed and accessible.
These recommendations are taken into account in the revised version of the manuscript
- Lastly, integrating the findings into the broader context of CNS regeneration research could enhance the manuscript. Discussing how the insights from salmon NSPCs contribute to the overall understanding of regeneration across species would underscore the study's relevance.
Thanks for the comment, these generalizing information have been added to the revised version of the manuscript.
- The manuscript does an excellent job of contrasting the regenerative capabilities of aNSPCs in fish (specifically zebrafish) and mammals. However, it could benefit from a deeper comparative analysis on how these differences in regenerative capacities might inform therapeutic strategies. Specifically, identifying key factors that enable zebrafish's high regenerative potential could offer insights into overcoming limitations in mammalian CNS regeneration.
Thanks for the comment, this information has been added to the Conclusion section.
Compared to mammals, where CNS regeneration is limited, zebrafish have demonstrated the ability to extensively regenerate even adult nerve tissues. This raises interest in the mechanisms underlying this difference. This is partly due to the presence of adult embryonic type cells in chum salmon, such as radial glia (Kroehne V, Freudenreich D, Hans S, Kaslin J, Brand M. Regeneration of the adult zebrafish brain from neurogenic radial glia-type progenitors. Development. 2011;138(22):4831-4841.) and the absence of these cells in the postembryonic state in mammals, as well as the number of NSCP zones in zebrafish and mammals. Thus, therapeutic strategies should focus on the possibility of activating these zones, searching for inhibitors or activators of neuroprotection factors and Wnt and Notch signaling pathways regulating neurogenesis. (Lindsey BW, Douek AM, Loosli F, Kaslin J. A whole brain staining, embedding, and clearing pipeline for adult zebrafish to visualize cell proliferation and morphogenesis in 3D. Front Neurosci. 2018; 12:546.)
(Ganz J, Kroehne V, Freudenreich D, Machate A, Geffarth M, Braasch I, et al. Subdivisions of the adult zebrafish pallium based on molecular marker analysis. F1000Res. 2014;3:308.)
- While discussing the behavior of aNSPCs and their role in regeneration, the manuscript could provide more details on the methodologies used to study these cells in both zebrafish and mammals. This includes specifics on intravital imaging, lineage analysis, and other experimental approaches that yield insights into aNSPC behavior, neurogenesis, and neuron integration into existing circuits.
This comment partially repeats paragraph 3. In accordance with the recommendations made, methodological and experimental schemes for the identification of NSPCs in mammals and zebrafish in trauma were added to the section "NSC in the adult brain".
- The discussion on the molecular markers of aNSPCs and the signaling pathways involved in their activation and proliferation is informative. Expanding on these molecular mechanisms, particularly those that differ between regeneratively competent and incompetent species, could enrich the narrative. Highlighting recent discoveries or ongoing debates in this area would also be beneficial.
Thank you for this comment, data on potential mechanisms in regenerative and incompetent species have been added to the article (sections « NSCP in the adult brain, the creation of ANSCS in intact and damaged brains»)
- The potential use of aNSPCs in regenerative therapy is a crucial aspect of this discussion. The manuscript might further explore the current barriers to using these cells in mammalian brain repair, including issues related to cell differentiation, integration, and functional recovery post-injury. Discussing ongoing research or experimental therapies that aim to overcome these barriers could provide a hopeful perspective on future advancements.
Potential problems of using NSCP in mammalian brain repair are the reactions of microglial cells and the inflammatory process, however, signaling pathways and neuroprotective factors can be used as a model for influencing not only fish, but also mammals
- The manuscript mentions neurogenic niches in both zebrafish and mammals, focusing on the SVZ and SGZ in mammals. Providing more detail on the characteristics, regulation, and functional significance of these niches in both species could offer readers a deeper understanding of the context in which aNSPCs operate. This could include discussions on the niche environment, cellular interactions, and how these factors influence neurogenesis.
Thank you for this comment. In accordance with the recommendations, data were added with more detailed characteristics, regulation and functional significance of SVZ and SGZ niches and other neurogenic zones described to date in the section "Creation of aNSCPs in intact and damaged brain"
- While the manuscript effectively reviews the current state of knowledge, it could more explicitly outline areas where further research is needed. This may include unanswered questions about the molecular pathways governing aNSPC behavior, challenges in translating findings from zebrafish to mammals, or the exploration of novel neurogenic regions outside the traditional niches.
Thank you for this comment, we consider this recommendation very useful and have added data on new neurogenic brain regions discovered in humans outside of traditional niches to the revised version of the manuscript.
- Ensure scientific terms are accurately used and fully explained where necessary. For example, when introducing molecular markers and signaling pathways, briefly describe their roles in neurogenesis to cater to readers who may not be familiar with these terms. Additionally, verifying that all references are current and relevant would strengthen the manuscript's credibility.
Thank you for this comment, all recommendations have been implemented and taken into account in the revised version of the manuscript.
- Including figures that visually summarize the differences in aNSPC characteristics, their localization in the brain, and their regenerative processes across species could enhance reader comprehension. Diagrams depicting the molecular pathways involved in aNSPC activation and differentiation could also be helpful.
Thanks for the comment, some new drawings have been added to the corrected version (Fig. 2), demonstrating the localization of doublecortin, an early marker of neurogenesis in different parts of the brain of adult trout, and Figure 5, illustrating some signaling pathways involved in the regulation of neurogenesis in the brain of fish.
- The manuscript does a commendable job highlighting the unique neurogenic capabilities of Pacific salmon. Expanding the comparative analysis with more vertebrate models, particularly those with limited regenerative capabilities, could further underscore the significance of the findings and help elucidate potential evolutionary pathways or mechanisms that have led to these differences.
Thanks for this comment, data on the peculiarities of the evolution of salmonids have been added to the section "Constitutive and reparative neurogenesis in the brain of Pacific salmon", as a result of which a full-genomic duplication of the genome occurred.
- While the document discusses the activation of NSPCs and their migration to injury sites, a more detailed exploration of the underlying molecular and cellular mechanisms would be beneficial. Specifically, identifying signaling pathways, transcription factors, and environmental cues that govern these processes in Pacific salmon could provide deeper insights into the basis of their regenerative prowess.
Thank you for this comment, in accordance with the recommendations made, a new scheme has been added to the text summarizing the influence of various factors on neurogenesis in the brain of fish. Also, data on the expression of various transcription factors Noth, Wnt, Shh, etc., Pax transcription factors, in particular, increased expression of Pax6 in the visual projection centers of the trout brain after unilateral eye damage (Pushchina, Varaksin, Neurolin expression in the optic nerve and immunoreactivity of Pax6-positive niches in the brain of rainbow trout (Oncorhynchus mykiss) after unilateral eye injury. Neural Regeneration Research 14(1):p 156-171, | DOI: 10.4103/1673-5374.243721), as well as a decrease in Pax2 expression in the telencephalon of juvenile chum salmon after TBI (Pushchina E.V., Zharikova E.I., Varaksin A.A. Mechanical brain injury increases cells’ production of cystathionine β-synthase and glutamine synthetase, but reduces Pax2 expression in the telencephalon of juvenile chum salmon, Oncorhynchus keta // International Journal of Molecular Sciences. 2021. Vol. 22, â„– 3. Articleno. 1279. doi.org/10.3390/ijms22031279).
The next information has been added to the section "Glial plasticity in response to disease and injury" and the Conclusion.
One of the key transcription factors are Sox2 and NeuroD, since zebrafish show high levels of expression during neurogenesis (Wegner M. All purpose Sox: The many roles of Sox proteins in gene expression. Int J Biochem Cell Biol. 2010;42(3):381-390.) (Ali F, Hindley C, McDowell G, Deibler R, Jones A, Kirschner M, et al. Cell cycle-regulated multi-site phosphorylation of Neurogenin 2 coordinates cell cycling with differentiation during neurogenesis. Development. 2011;138(19):4267-4277.)
The Wnt/β-catenin and Notch signaling pathways are involved in NSCP differentiation and regulation of neurogenesis
(Jorstad NL, Wilken MS, Grimes WN, Wohl SG, VandenBosch LS, Yoshimatsu T, et al. Stimulation of functional neuronal regeneration from Müller glia in adult mice. Nature. 2017;548(7665):103-107.) (Andersson ER, Salto C, Villaescusa JC, Cajanek L, Yang S, Bryjova L, et al. Wnt5a cooperates with canonical Wnts to generate midbrain dopaminergic neurons in vivo and in stem cells. Proc Natl Acad Sci U S A. 2013;110(7):E602-610.)
In addition, pathways involved in the regulation of the cell cycle, such as cyclin-dependent kinase (CDK) and cyclin-dependent kinase inhibitors (CKI), are closely related to the processes of neuronal differentiation (Pilaz LJ, Silver DL. Post-transcriptional regulation in corticogenesis: how RNA-binding proteins help build the brain. Wiley Interdiscip Rev RNA. 2015;6(5):501-515.)
Mitochondrial pathways are also important in neurogenesis in fish. Activation levels of mitochondrial factors such as PGC-1a can affect the energy metabolism of neuronal stem cells and their ability to differentiate. (St-Pierre J, Drori S, Uldry M, Silvaggi JM, Rhee J, Jäger S, et al. Suppression of Reactive Oxygen Species and Neurodegeneration by the PGC-1 Transcriptional Coactivators. Cell. 2006;127(2):397-408.) (Hom JR, Quintanilla RA, Hoffman DL, de Mesy Bentley KL, Molkentin JD, Sheu SS, et al. The Permeability Transition Pore Controls Cardiac Mitochondrial Maturation and Myocyte Differentiation. Dev Cell. 2011;21(3):469-478.)
Transcription factors, signaling and cellular pathways represent a complex network of interacting elements that regulate neurogenesis in fish. The study of these molecular mechanisms opens up new prospects for the development of strategies to stimulate neurogenesis and, possibly, the treatment of neurological disorders in humans.
- The manuscript could benefit from additional information on the experimental approaches used to study NSPCs in Pacific salmon, including techniques for tracking cell proliferation, migration, and differentiation in vivo. Such details would help readers appreciate the complexity of the research and the validity of the conclusions drawn.
Thanks for this comment, in accordance with the recommendations made, data on transgenic studies were added to the corrected version (Pushchina E.V., Kapustyanov I.A., Varaksin A.A. Viral Vectors in Transgenic Research: Prospects for the Treatment of CNS Diseases and Gene Therapy. Pacific Medical Journal. 2022;(1):46-55. ( In Russ.) https://doi.org/10.34215/1609-1175-2022-1-46-55 ), allowing to track the proliferation, neurogenic differentiation of cells of the mesencephalic tegmentum of juvenile chum salmon (Pushchina, E.V.; Kapustyanov, I.A.; Shamshurina, E.V.; Varaksin, A.A. A Confocal Microscopic Study of Gene Transfer into the Mesencephalic Tegmentum of Juvenile Chum Salmon, Oncorhynchus keta, Using Mouse Adeno-Associated Viral Vectors. Int. J. Mol. Sci. 2021, 22, 5661. https://doi.org/10.3390/ijms22115661 ), as well as estracerebellar projections of juvenile chum salmon (Pushchina, E.V.; Bykova, M.E.; Shamshurina, E.V.; Varaksin, A.A. Transmission of Brain Neurons in Juvenile Chum Salmon (Oncorhynchus keta) with Recombinant Adeno-Associated Hippocampal Virus Injected into the Cerebellum during Long-Term Monitoring. Int. J. Mol. Sci. 2022, 23, 4947. https://doi.org/10.3390/ijms23094947).
- Discussing the functional integration of newly formed neurons into existing neural circuits and their impact on the behavior or physiological functions of Pacific salmon would enhance the manuscript. This aspect is crucial for understanding the biological significance of the observed neurogenic processes, particularly in the context of the salmon's life cycle and adaptation to environmental changes.
Thanks for this comment, in accordance with the recommendations made, the information was added to the section "Constitutive and reparative neurogenesis in the brain of Pacific salmon".
- The manuscript hints at the potential implications of salmon NSPCs for regenerative medicine. Delving into specific strategies for translating these findings into mammalian models, including humans, could make the discussion more relevant to the broader field of neurodegenerative disease research and treatment development.
This question is identical to paragraph 5. Please see the answer to paragraph 5
- The mention of NSPCs in older individuals introduces an interesting aspect of age-related changes in neurogenic capacity. Expanding on how the properties of NSPCs and their regenerative potential change with age, and the factors influencing these changes, would add depth to the discussion on the plasticity and resilience of the salmon brain.
Thank you for this comment, indeed, age-related changes in regenerative potential are associated with a number of factors, the discussion of which we briefly presented in the Conclusion
- The manuscript would benefit from a clearer articulation of future research directions, particularly addressing the gaps in understanding the regulation of NSPCs, their long-term functional integration, and the possibility of harnessing these cells for therapeutic purposes. Highlighting specific questions and potential experimental approaches would guide future studies in this exciting area of research.
Thank you for Your comment, the following additions have been made to the Conclusion section:
Activation of brain stem cells in fish is accompanied by a change in the expression of molecular markers such as Sox2, NeuroD,(Zupanc GKH, Sîrbulescu RF. Adult Neurogenesis and Neuronal Regeneration in the Central Nervous System of Teleost Fish. Eur J Neurosci. 2011;34(6):917-929.) ((Dyatel N, BeilT, Strähle U, Rastegar S. Differential expression of id genes and their potential regulator znf238 in zebrafish adult neural progenitor cells and neurons suggests distinct functions in adult neurogenesis. Gene Expr Patterns. 2015;19(1-2):1-13.) and other transcription factors, so the use of pharmacological agents that affect their activation can improve the regenerative potential of the brain.
- The comparison between zebrafish and mammals offers a valuable perspective on the evolutionary conservation of astroglial functions. To deepen this analysis, consider discussing the implications of these similarities and differences for our understanding of astroglia's roles across the animal kingdom. Highlighting specific evolutionary advantages or adaptations in astroglial functions could provide intriguing insights.
Thank You for this comment. In accordance with the recommendations made, data on the features of the immune post-traumatic response in regenerator-competent organisms (danios) and in mammals were added to the section "Heterogeneity of stem cell pools and radial glia". Additionally, various aspects of the course of the immune response in these groups of vertebrates, its effectiveness and impact on the regeneration process are discussed.
- While the section effectively outlines the impact of astroglia on neural activity and behavior, it could benefit from more detailed explanations of the underlying mechanisms. For example, how do astroglial cells detect neuronal activity or environmental changes, and what are the signaling cascades involved in their response? Expanding on these processes would enrich the narrative.
Thank you for this comment. Considering that the data on the effect of radial glia on the behavioral activity of danio are currently quite limited, we have slightly expanded this section with some details that have been studied to date.
- The manuscript mentions the use of optogenetics and other methodologies to study astroglial-neuronal interactions. Delving into these techniques, their advantages, and limitations could offer readers a clearer understanding of how current findings were obtained and what future studies might look like. Discussing the potential of emerging technologies, such as in vivo imaging and single-cell transcriptomics, could also be valuable.
Thank You for this comment, this recommendation has been taken into account, new information has been added to the text. In particular, thanks for your comment. In a study by Gerasimov et al. The use of optogenetic methods to stimulate mouse hippocampal astrocytes in order to modulate the activity of neurons is shown. The results obtained are useful for future experimental evaluation of astrocyte activation in the context of models of Alzheimer's disease and other neurodegenerative diseases.
- Expanding on the functional implications of astroglial-neuron interactions in both physiological and pathophysiological states would enhance the section. Specifically, how do these interactions contribute to the regulation of complex behaviors or the development of neurological disorders? Addressing these questions could highlight the clinical relevance of the research.
Thanks for Your comment. Takahashi's work is devoted to the role of astroglia in brain metabolism and regulation of cerebral blood flow in norm and pathology, the importance of astroglial-neuronal interactions, as well as the relevance of astroglia research for the development of therapeutic strategies for the treatment of neurological diseases.
- The discussion on gap junctions is particularly intriguing. Further elaboration on how gap junction connectivity influences the coordination of astroglial and neuronal networks during different brain states or in response to injury would be beneficial. Additionally, exploring the potential for targeting gap junctions in therapeutic strategies for neurological conditions could be interesting.
Thanks for Your comment. In the work of Chever et al. astrocytic networks, the effect of astrocytes on neuronal activity, the relationship of the pathological state of astrocytic networks with the occurrence of epileptiform events and convulsive activity are studied, and the prospects of these studies for the development of anticonvulsant therapy methods are considered.
- The section concludes with an optimistic outlook on future research using zebrafish as a model organism. Articulating specific questions or hypotheses that future studies should address would be helpful. For instance, how might astroglia contribute to the resilience or vulnerability of neural circuits to injury or disease? What are the potential therapeutic targets within astroglial signaling pathways?
Thank You for your comment, these recommendations have been taken into account and added to the final section of the article.
- A deeper dive into how astroglial functions are tailored to the specific physiological needs or environmental challenges faced by different species could offer fascinating evolutionary insights. For example, how might the astroglial response to neurotransmitters like norepinephrine or acetylcholine differ between species with vastly different lifestyles or habitats?
Thank You for your comment, these recommendations have been taken into account and added to the final section of the article.
- Briefly touching upon the role of astroglia in neurodevelopment and how their functions evolve from embryonic stages to adulthood could provide a more comprehensive view of their importance in the CNS.
Thank You for your comment, these recommendations have been taken into account and added to the final section of the article.

Reviewer 2 Report
Comments and Suggestions for Authors
The manuscript presents an interesting topic regarding the properties of neural stem progenitor cells in adult teleost fishes and makes a parallel between them and mammals. The regeneration of the nervous system represents a challenge and a better understanding of the plasticity of NCPCs in vertebrate species where this regeneration is possible, such as fish, can improve therapeutic perspectives in mammals, respectively humans. The authors have previous studies in the addressed field and make a comprehensive and well-documented review.
The figures are well made and suggestive.
The manuscript brings also the glial cells and neuroepithelial cells to the fore.
The chapters are well structured and touch on most of the key points regarding adult neurogenesis of teleost fish. However, a final chapter that draws some conclusions at the end of all the data presented is necessary.
Author Response
Thanks for the recommendations, in accordance with the expressed wishes, a final chapter Conclusion has been added to the work, which summarizes and presents the resulting conclusions on all the data presented.

Reviewer 3 Report
Comments and Suggestions for Authors
The authors have provided a comprehensive and up-to-date overview of the mechanisms of brain development and regeneration in teleost fish. They have also compared these mechanisms with those of vertebrates, highlighting the differences in brain regeneration processes and pointing out possible implications for regenerative medicine.
An important addition to this review would be the inclusion of illustrative material showing the presence of stem/progenitor cells in the brain of teleost fish compared to other model organisms such as laboratory rodents. In addition, a chapter discussing approaches to studying stem and progenitor cells in teleost fish, zebrafish and laboratory mice would be beneficial. The authors should also characterize the conservatism of neural stem/progenitor cell markers to determine if common antibodies can be used across species. This would be useful for researchers who are working on this issue on one species and would like to expand their research to another species using the developed methodological base.
Minor issue
It is necessary to add licensed images of Biorender with an appropriate resolution of 300 dpi.
It is necessary to identify all cell types in Figures 2 and 3.
Author Response
The authors have provided a comprehensive and up-to-date overview of the mechanisms of brain development and regeneration in teleost fish. They have also compared these mechanisms with those of vertebrates, highlighting the differences in brain regeneration processes and pointing out possible implications for regenerative medicine.
Thank you for this comment, an additional scheme has been added to the article, as well as an installation with doublecortin-expressing cells in various areas of the brain of adult trout.
An important addition to this review would be the inclusion of illustrative material showing the presence of stem/progenitor cells in the brain of teleost fish compared to other model organisms such as laboratory rodents. In addition, a chapter discussing approaches to studying stem and progenitor cells in teleost fish, zebrafish and laboratory mice would be beneficial. The authors should also characterize the conservatism of neural stem/progenitor cell markers to determine if common antibodies can be used across species. This would be useful for researchers who are working on this issue on one species and would like to expand their research to another species using the developed methodological base.
Data discussing approaches to the study of adult neurogenesis in fish and mammalian models have been added to the revised version of the work.
This is an interesting and profound recommendation. According to generally accepted approaches that have been applied to both fish and mammals, a number of molecular markers such as can be used as universal markers of NSCP.
It is necessary to add licensed images of Biorender with an appropriate resolution of 300 dpi.
This recommendation has been taken into account, and Biorender images with an appropriate resolution of 300 dpi have been added to the corrected version.
It is necessary to identify all cell types in Figures 2 and 3.
This recommendation is taken into account in Fig. 2 and 3 added information about cell morphotypes.

Round 2
Reviewer 1 Report
Comments and Suggestions for Authors
I wish to extend my heartfelt thanks to the authors for their thorough and meticulous revisions. The manuscript has significantly improved since its first submission, reflecting the authors' earnest dedication to addressing the previously highlighted issues.
Comments on the Quality of English LanguageMinor editing of English language required